# Transport of the 2017 Canadian wildfire plume to the tropics via the Asian monsoon circulation

Corinna Kloss[1], Gwenaël Berthet[1], Pasquale Sellitto[2], Felix Ploeger[3], Silvia Bucci[4], Sergey Khaykin[5], Fabrice Jégou[1], Ghassan Taha[6], Larry W. Thomason[7], Brice Barret[8], Eric Le Flochmoen[8], Marc von Hobe[3], Adriana Bossolasco[1], Nelson Bègue[9], and Bernard Legras[4]

[1]Laboratoire de Physique et Chimie de l'Environnement et de l'Espace, CNRS/Université d'Orléans, UMR 7328, Orléans, France
[2]Laboratoire Interuniversitaire des Systèmes Atmosphériques, UMR CNRS 7583, Université Paris-Est Créteil, Université de Paris, Institut Pierre Simon Laplace (IPSL), Créteil, France
[3]Forschungszentrum Jülich GmbH, Institute of Energy and Climate Research (IEK-7), Jülich, Germany
[4]Laboratoire de Météorologie Dynamique, UMR 8539, CNRS – École Normale Supérieure/Université Pierre et Marie Curie/École Polytechnique, Paris, France
[5]LATMOS, Université Versailles St-Quentin, Sorbonne Université, CNRS, IPSL, Guyancourt, France
[6]Universities Space Research Association, Columbia, Maryland, USA
[7]NASA-Langley Research Center, Hampton (VA), USA
[8]Laboratoire d'Aérologie, Université de Toulouse, CNRS, UMR 5560, UPS, Toulouse, France
[9]Laboratoire de l'Atmosphère et des Cyclones, CNRS, UMR 8105, Université de la Réunion, France

**Correspondence:** Corinna Kloss (corinna.kloss@cnrs-orleans.fr)

**Abstract.** We show that a fire plume injected into the lower stratosphere at high northern latitudes during the Canadian wildfire event in August 2017 partly reached the tropics. The transport to the tropics was mediated by the anticyclonic flow of the Asian monsoon circulation. The fire plume reached the Asian monsoon area in late August/early September, when the Asian Monsoon Anticyclone (AMA) was still in place. While there is no evidence of mixing into the center of the AMA, we show that a substantial part of the fire plume is entrained into the anticyclonic flow at the AMA edge, and is transported from the extra-tropics to the tropics, and possibly the Southern Hemisphere particularly following the north-south flow on the eastern side of the AMA. In the tropics the fire plume is lifted by $\sim$5 km in 7 months. Inside the AMA we find evidence of the Asian Tropopause Aerosol Layer (ATAL) in August, doubling background aerosol conditions with a calculated top of the atmosphere shortwave radiative forcing of -0.05 W/m$^2$. The regional climate impact of the fire signal in the wider Asian monsoon area in September exceeds the impact of the ATAL by a factor of 2-4 and compares to that of a plume coming from an advected moderate volcanic eruption. The stratospheric, trans-continental transport of this plume to the tropics and the related regional climate impact point at the importance of long-range dynamical interconnections of pollution sources.

## 1 Introduction

During the 2017 summer season, historically severe wildfires appeared in western Canada and in the north-western United States. Strong thunderstorms (pyro-cumulonimbus activity), which developed above the fires, injected smoke particles above western Canada into the lower extratropical stratosphere in mid-August (Khaykin et al., 2018). The fire plume was transported

through the jet stream eastward and rose 2-3 km per day within the first days after its injection into the stratosphere (Khaykin et al., 2018) and to an altitude of 23 km within two months (Yu et al., 2019). Three days after the first appearance in the stratosphere above Canada, the plume first appeared over Europe on August $19^{th}$. Above southern France the plume was observed at altitudes up to about 20 km (Khaykin et al., 2018). Multiple studies have analyzed the fire plume above western/central Europe

with LiDAR observations (Khaykin et al., 2018; Ansmann et al., 2018; Peterson et al., 2018). Yu et al. (2019) recently showed with model calculations that most of the fire plume in the stratosphere was quickly transported to the poles (depending on the computed BC content). The general impact on the radiative balance and climate of aerosol plumes from wildfires in the lowermost stratosphere has been discussed in Ditas et al. (2018); they found that the global average direct radiative forcing at the top of the atmosphere (TOA) of biomass burning aerosols from wildfires may reach -0.20 W/m$^2$ (including biomass burning

plumes and biomass burning-affected background atmosphere, and including absorbing and scattering aerosol components). A few rare extreme-fire events prior to this case have been investigated in terms of global distribution, showing enhanced trace gas and aerosol signatures in tropical latitudes (e.g. Siddaway and Petelina (2011), Pumphrey et al. (2011), Jost et al. (2004) and Fromm et al. (2008)). Those studies primarily focus on the evolution and distribution of the respective fire plumes rather than the underlying transport processes.

The Asian summer monsoon influences the composition of the Upper-Troposphere–Lower-Stratosphere (UTLS) (Garny and Randel, 2016; Pan et al., 2016; Ploeger et al., 2017), especially in the Tropical Tropopause Layer (TTL: as defined by Fueglistaler et al. (2009), see their Fig. 5a). Either within the upwelling of the Asian monsoon or through the dynamical transport around the Asian Monsoon Anticyclone (AMA), air masses are drawn up each year between approximatively June and September into the TTL. It has been shown that most of the air entering the stratosphere in the tropics is transported around

the AMA circulation without entering its core (Tissier and Legras, 2016). Within the AMA, air masses of continental origin are likely trapped, due to the dynamical barrier of the surrounding jets and enhanced tropospheric and reduced stratospheric tracer concentrations are typically observed inside the transport barrier (Park et al., 2008; Randel et al., 2010; Bergman et al., 2013; Santee et al., 2016). Additionally, an accumulation of aerosols has been found inside the AMA, the Asian Tropopause Aerosol Layer (ATAL) (Vernier et al., 2011; Yu et al., 2015). The ATAL existence has been attributed to the recent increase of

Asian emissions of anthropogenic pollutants like sulphur dioxide and volatile organic compounds (R. Neely et al., 2014; Yu et al., 2015) and is sustained by the convective activity of the Asian monsoon. The composition, variability, trend and budget of the ATAL are largely uncertain and are currently studied. The impact of the ATAL on the extratropical aerosol budget in the northern hemisphere (NH) has been investigated by Khaykin et al. (2017). On average, the geographical extent of the AMA is largest in mid-July through beginning of August, then decreases until it completely dissipates by the end of September (Santee

et al., 2016). Lagrangian transport simulations suggest that 5% of the air mass in the tropical pipe at 460 K and 15% in the extratropical lowermost stratosphere at 380 K originate in the anticyclone over the course of a year (Ploeger et al., 2017). Conditions of enhanced aerosol concentration such as volcanic eruptions can act as tracers of AMA dynamics. For the Nabro volcano eruption, for example, the emitted aerosol and precursors have been partly injected directly into the lower stratosphere (Vernier et al., 2013; Fromm et al., 2013) at altitudes of about 15-18 km (Clarisse et al., 2014; Fromm et al., 2014). It has been

shown that the Asian monsoon anticyclone was a dominant feature governing the dispersion of this volcanic plume (Fairlie

et al., 2014). Satellite observations of volcanic effluents as $SO_2$ (Clarisse et al., 2014) and sulphate aerosols (Sellitto et al., 2017) have shown the interaction of the Nabro plume's horizontal dispersion and the AMA dynamics. During the year of the Sarychev eruption, which occurred at the beginning of the Asian monsoon season, a negative anomaly inside the AMA has been shown, while for all other years, there is a positive aerosol anomaly (ATAL) (Vernier et al., 2011). Both observations

(for the Sarychev and Nabro eruption) support a transport barrier that separates 'inside' and 'outside'. Volcanic aerosol plumes remaining outside of the AMA (as for the Sarychev eruption) can instead be transported along the eastern flank of the AMA circulation into the tropics (Wu et al., 2017). The importance of the Asian monsoon circulation for the horizontal transport of extratropical air masses into the tropics has already been shown by Konopka et al. (2010).

The Canadian fire event of August 2017 occurred during the same timeframe of the Asian monsoon season in 2017. Here, we

investigate the impact of the AMA circulation on the horizontal transport of the fire plume from the Canadian fires in 2017. In addition, we estimate the regional climate impact of the fire plume in the Asian monsoon area and we compare this impact to the one of the ATAL during the monsoon season. Furthermore, the wider impact of the plume is compared with that of a moderate volcanic eruption.

The paper is structured as follows: data sets and methods are described in Section 2. The pathway of the fire plume to the Asian

monsoon region is analyzed in Section 3. The corresponding regional climate impact is estimated in Section 4. Conclusions are drawn in Section 5.

## 2 Methods

### 2.1 SAGE III-ISS aerosol extinction profiles

Stratospheric Aerosol and Gas Experiment (SAGE III-ISS) was launched in February 2017 and mounted aboard the Interna-

tional Space Station. Data from the instrument is available from June 2017 onwards. SAGE III uses solar and lunar occultation and limb-scatter to infer profiles of trace gases like ozone and aerosol extinction coefficient at nine wavelengths between 384 and 1544 nm. Due to orbital considerations, SAGE III acquires 30 sets of profiles per day in two latitudes bands which span from roughly 60°N to 60°S over the course of a month with best spatial coverage in the mid-latitudes (30-60°). The vertical extent of the aerosol extinction coefficient profiles is from roughly 40 km down to encountering either the solid Earth or ex-

ceeding the dynamical range of the detector usually in association with an optically opaque water cloud. These profiles have a vertical resolution of ∼1 km and are reported every 0.5 km from 0.5 to 40 km. The horizontal resolution is approximately 200 km along the line of sight between the instrument and the Sun and stretched along the direction of motion of the platform (ISS) by an additional 200 km. The meteorological data including the height of the tropopause are derived from the Modern-Era Retrospective analysis for Research and Applications, Version 2 (MERRA-2). Herein, we make use of the cloud-unfiltered

version 5.1 solar occultation aerosol extinction coefficient profiles at 3 wavelengths (384, 521, 676 nm).

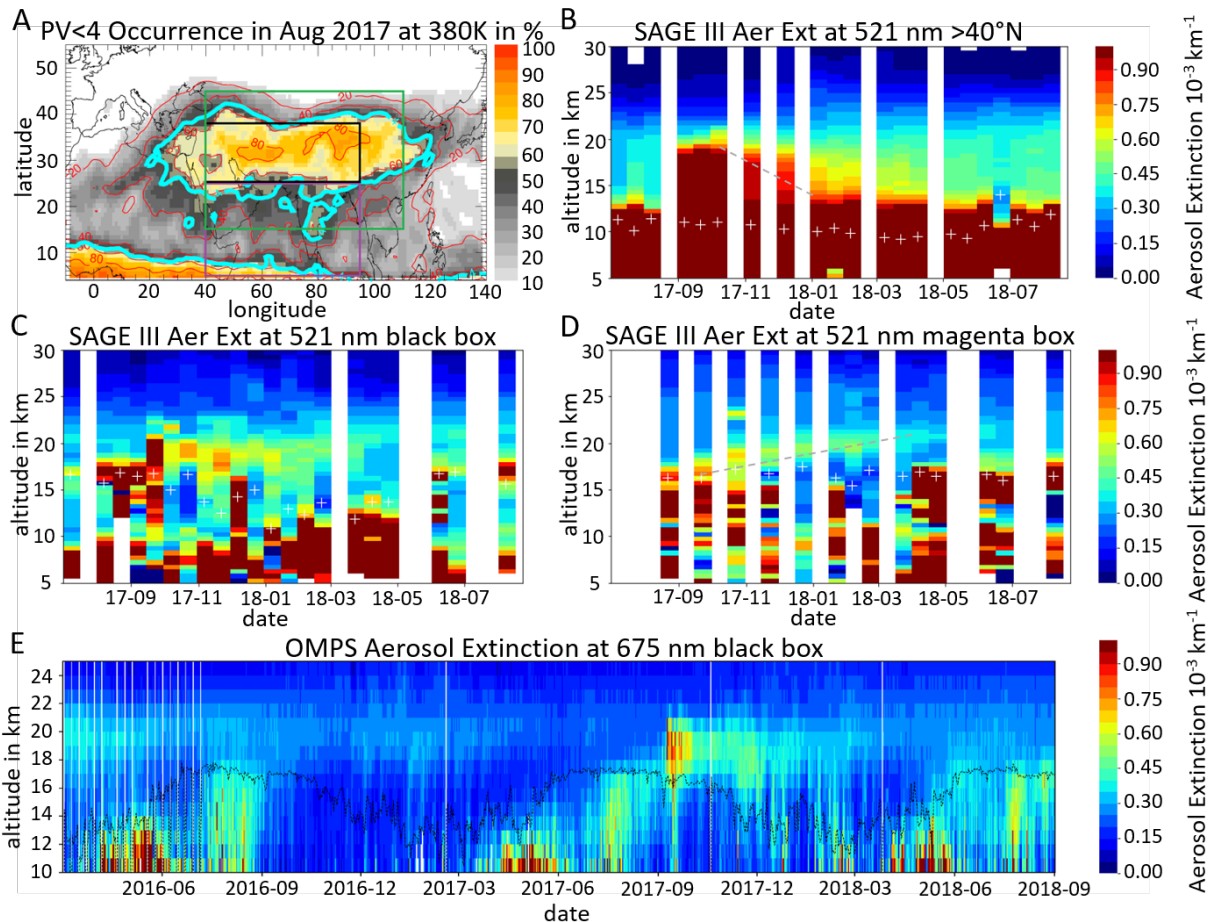

**Figure 1.** (A) Selection criteria for the defined AMA box (black box, 25°N-38°N and 40°E-95°E) for this paper, a southern area (magenta box, 5°N-25°N and 40°E-95°E), the wider Asian monsoon area box (green box: 15°N-45°N and 40°E-110°E): the occurrence frequency of the AMA at 380K (color filling) in percentage of days for August $1^{st}$ to September $5^{th}$ in 2017 is used as a criterion for the selection of the boxes. It is based on the probability of a grid point having a PV value below 4 PVU, according to Ploeger et al. (2015). Red contours represent the percentage values; the cyan contour shows the average PV transport barrier. (B), (C) and (D) SAGE III aerosol extinction values at 521nm on 15 day averages for (B) all longitudes and latitudes available above 40°N with an average of 88 profiles per bin (however, never higher than 70°N), (C) for the black box in A, with an average of 5 profiles per bin, (D) for the magenta box in A with an average of 7 profiles per bin. White plus symbols represent the average tropopause altitude of the averaged profiles. Grey dashed lines in (B) and (D) represent the observed descent/ascent. (E) OMPS daily median aerosol extinction measurements in the black box at 675 nm. The black line represents the daily mean tropopause height in the black box.

## 2.2 OMPS aerosol extinction profiles

The Ozone Mapping Profiler Suite Limb Profiler (OMPS-LP) instrument onboard Suomi National Polar-orbiting Partnership (Suomi NPP) images the Earth limb by pointing aft along the spacecraft flight path. The sensor employs 3 vertical slits separated horizontally to provide near global coverage in 3-4 days. The instrument measures limb scattering radiance and solar irradiance at the 290 – 1000 nm wavelength range and the 0-80 km altitude range with 1.6 vertical resolution. The current OMPS-LP algorithm uses the radiance measurements at a single wavelength (675 nm) to estimate the aerosol extinction coefficient profile (Loughman et al., 2018; Deland, 2016). In this study, we use the OMPS version 1.5 aerosol extinction data (Chen et al., 2018),

left slit only, from May 2016 to 2018 in the area 25-38°N and 40-95°E. Data are provided between 10 and 40 km altitude. Clouds have been filtered by removing the data below the cloud top (Chen et al., 2016) for clouds below the tropopause. Hence, potential biomass, volcanic and fire plume aerosol signals are maintained in the stratosphere. However, some residual cloud contamination may remain. As for SAGE III, the tropopause altitude is provided by MERRA-2 forward processing.

## 2.3 IASI CO observations

To visualize the fire plume in an early stage, when OMPS gives no measurement due to oversaturation, the Infrared Atmospheric Sounding Interferometer (IASI) CO data are chosen as a representation of its position. IASI is a thermal infrared nadir sensor flying on board the MetOp A, B and C satellites launched in 2006, 2012 and 2016, respectively. The IASI series enables monitoring the tropospheric content of atmospheric trace gases such as $O_3$ (Eremenko et al., 2008; Barret et al., 2011) and CO (George et al., 2009; De Wachter et al., 2012) with a global coverage twice daily, for each individual sensor. Here, we use CO

data retrieved with the Software for a Fast Retrieval of IASI Data (SOFRID) (Barret et al., 2011; De Wachter et al., 2012). Validation against MOZAIC airborne in-situ data have demonstrated that SOFRID-CO data are able to capture the seasonal variability of CO at mid-latitudes (Frankfurt) as well as at tropical latitudes (Windhoek) in the lower (and upper) troposphere with correlation coefficients of 0.85 (and 0.70 respectively) (De Wachter et al., 2012). Barret et al. (2016) have also shown that SOFRID-CO data were able to document the accumulation of CO in the UTLS AMA.

## 2.4 CLaMS CO simulations

The Chemical Lagrangian Model of the Stratosphere (CLaMS) is a Lagrangian Chemistry transport model. The transport is modelled based on 3D forward trajectories and an additional parameterization of small-scale mixing (McKenna et al., 2002; Konopka et al., 2007). For this study, the model transport is driven by ERA-Interim meteorological data (Dee et al., 2011). Here, the CO trace gas enhancements in the AMA simulated by CLaMS are used to confirm the location and isolation of the

anticyclone indicated by the PV field. The lower boundary condition are derived from Atmospheric Infrared Sounder (AIRS) version 6 satellite measurements following the approach presented by Pommrich et al. (2014). Chemical loss is included in the model based on reactions with OH (Pommrich et al., 2014). However, it has to be noted that upward transport in the troposphere is likely too weak in ClaMS due to the lack of a convective parameterization and therefore, the simulated CO mixing ratio in the Asian monsoon area are likely too low. However, the simulated CO indicates the isolation and position of the AMA.

## 2.5 TRACZILLA back-trajectories

Back-trajectories are computed using archived horizontal winds and vertical velocities from the European Centre for Medium-Range Weather Forecasts (ECMWF) reanalyses (Pisso and Legras, 2008). For trajectories remaining in the Asian area, ERA-5 kinematics are used for the calculation for 1-hourly time steps with a horizontal resolution of 0.125° x 0.125° and a vertical resolution of ∼300 to 500 m. For back trajectories leaving Asia, the global Era-Interim input is used (Dee et al., 2011), with a vertical resolution of around 1 km at the considered altitudes and a 1°x1° horizontal resolution on a global scale and 3-hourly time steps. The time, pressure level and region of the air parcels release is chosen according to the individual SAGE III profiles of interest. Here, 1000 trajectories are initiated on a regular grid covering the spatial uncertainty of the SAGE III measurements.

## 2.6 UVSPEC Radiative forcing estimations

The shortwave surface and TOA direct radiative forcing for this event is estimated using the UVSPEC radiative transfer model and the LibRadtran package (Mayer and Kylling, 2005), available at the following website: http://www.libradtran.org/doku.php. The radiative transfer equation is solved with the SDISORT method, the pseudo-spherical approximation of the discrete ordinate method (DISORT) (Dahlback and Stamnes, 1991). Surface and TOA direct and diffuse shortwave spectra are computed in the range 300.0 to 3000.0 nm (0.1 nm spectral resolution). We use the input solar flux spectra of Kurucz (2005). The atmospheric state in terms of the vertical profiles of temperature, pressure, humidity and gas concentration is set as for the AFGL (Air Force Geophysics Laboratory) climatological standard tropical atmosphere (Anderson et al., 1986). Molecular absorption is parameterized with the LOWTRAN band model (H. Pierluissi and S. Peng, 1985), as adopted from the SBDART code (Ricchiazzi et al., 1998). We performed clear-sky simulations. We perform a baseline simulation, with this setup and a background boundary layer aerosol layer. Then we add the measured fire plume and ATAL extinction coefficient profiles (SAGE III observations). For baseline and fire plume- or ATAL-perturbed configurations, we simulate the radiative transfer at different solar zenith angles (SZA). The daily average shortwave TOA radiative forcing for a given aerosol layer (fire plume or ATAL) is calculated as the SZA-averaged upward diffuse irradiance for a baseline simulation without the investigated aerosols minus that with aerosols, integrated over the whole spectral range. The shortwave surface radiative forcing is calculated as the SZA-averaged downward global (direct plus diffuse) irradiance with aerosols minus baseline, integrated over the whole spectral range.

## 3 Tracing of the fire plume in the Asian monsoon region

To study the transport of the Canadian fire plume in the Asian monsoon area, we distinguish two sub-regions: $1^{st}$ the main Asian monsoon region 15-45°N and 40-110°E (the green box in Fig. 1A), a compromise from the choices of Santee et al. (2016), Ploeger et al. (2015), Vernier et al. (2011) and $2^{nd}$ the AMA box in 2017 25°N-38°N and 40°E-95°E, with maximized probability of being inside the AMA at 380 K for August 2017 (the black box in Fig. 1A). This probability (occurrence frequency) has been calculated according to a maximum in the PV gradient on the 380K isentrope following Ploeger et al.

(2015). Even though this is considered as a reliable method to identify the AMA center, it should be noted that the maximum in the PV gradient around the monsoon anticyclone is weak (see discussion in Ploeger et al. (2015)) and has even not been detected for several days during summer 2017. Based on the strength of the easterly jets in 2017 the circulation of the anticyclone was strongest during July and from thereon declined very slowly and was still in place, even if weaker, at the end of September (see Fig. A1).

Fig. 1B shows the aerosol extinction values for all longitudes available in the SAGE III above 40°N. To exclude most cloud features and also background aerosol in Fig. 1B, we focus on the aerosol extinction region from 0.6-0.9 $10^{-3}$ km$^{-1}$ for our analysis. A strongly enhanced aerosol extinction signature appears in the SAGE III data set, >40°N mid to end of August (Fig. 1B), after the beginning of the major fire event in Canada. This confirms the results of multiple previous studies, which have also seen highly increased aerosol signatures due to the fire mid to end of August at higher latitudes in the NH (Khaykin et al., 2018; Ansmann et al., 2018; Haarig et al., 2018). It has to be noted that the measurement point furthest north was taken below 70°N, however, especially in the winter most observations are limited to below 50°N. Between November 2017 and March 2018, the aerosol signature descends with roughly 0.64 mm in altitude per second based on aerosol extinction values > 0.8 $10^{-3}$ km$^{-1}$ (5 km in three months, October to January). This is in the order of the rate expected for the downwelling of the BDC (see Abalos et al. (2015)). However, other processes may contribute to the observed aerosol extinction decrease in Fig 1B: the effect of sedimentation is expected to play an important role. The contribution of sedimentation as well as dilution/mixing is not quantified here, because additional microphysical and dynamical sensitivity studies would be necessary. The lower stratosphere is filled with enhanced aerosols until about mid-April 2018. Fig. 1C shows the SAGE III aerosol extinction values in the inner AMA region (black box in Fig. 1A). The unfiltered cloud structures in the SAGE III data set masks the first appearance of the plume in the back box in Fig. 1C. However, the first SAGE III profile in the Asian monsoon region, that we can trace back to the fire plume signature and that has previously been transported within the circulation of the AMA appears on August 30$^{th}$ 2017 at ~17 km altitude (Fig. B1, within the green box). As a result, there is no indication of a fire plume signature within the AMA core (i.e. passing the AMA transport barrier). The relatively high altitudes of this signature (17-20 km) in Fig. 1C indicate that the fire plume arrived in the TTL region in the Asian monsoon area, where the transport barrier might have forced the fire plume to rise by bypassing the AMA on its upper part, as it was the case for the Sarychev aerosol plume in 2009 (Vernier et al., 2011). A clear signal is still apparent in April 2018, 8 months after its first appearance and long after the break down of the AMA confinement. However, it has to be noted that there are no previous years of SAGE III measurements available so that no comparison with background conditions in April can be made.

To see whether the fire plume has been transported to the tropics (as it has been shown for the Sarychev eruption by Wu et al. (2017)), another box south of the core Asian monsoon box has been chosen (Fig. 1D, according to Fig. 1A, magenta box). We attribute the ascending signal starting at around 16 km in mid September and reaching altitudes of around 21 km about 6 months later to the Canadian wildfire, as its origin coincides in time and altitude with the fire signal in the AMA region (black box, Fig. 1C). Hence, the AMA generates a strong dynamical connection between the mid-latitudes and the tropics during the summer season. In the tropics, the fire plume signature rises about 0.2-0.3 mm per second (about 5 km from September to April) in the magenta box according to aerosol extinction values of around 0.6 $10^{-3}$ km$^{-1}$. This tropical upwelling velocity estimate

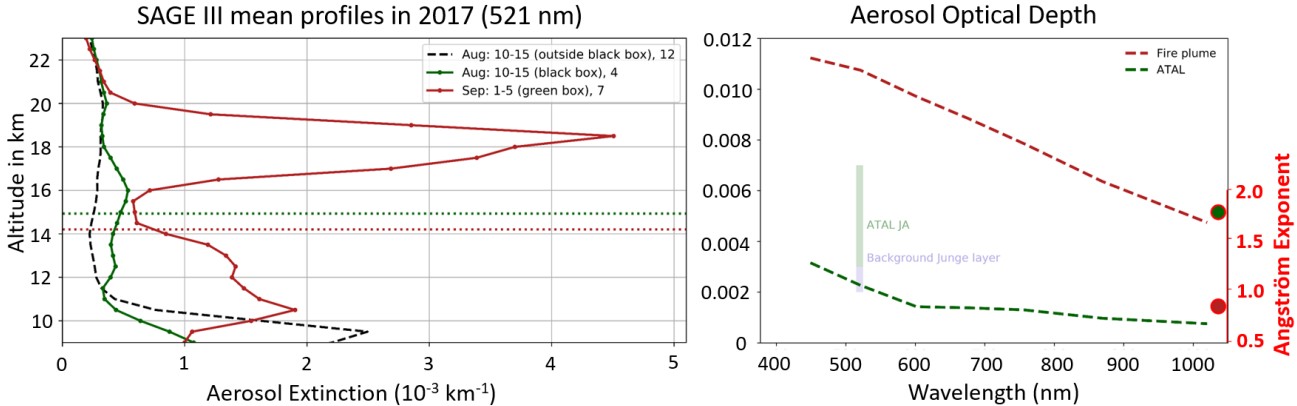

**Figure 2.** (A) Mean SAGE III aerosol extinction profiles in the UTLS at 521 nm in defined areas: the green curve shows the average for the black box from Fig. 1A, from August $10^{th}$ to $15^{th}$. The black curve is the equivalent mean extinction profile for the area 15-45°N and 60°W-10°E. The red curve shows the average for the green box in Fig. 1A from $1^{st}$-$5^{th}$. Tropopause heights, given by the SAGE III data set are indicated (dotted green line: ±2.7 km, dotted red line: ±2.3 km). Number of profiles averaged are indicated. (B) Spectral AOD from SAGE III observations for the mean fire and ATAL (dashed red and green lines respectively). The interval of CALIOP AODs at 550 nm for July-August (JA) ATAL and Junge layer are also displayed (semi-transparent green and blue, respectively) Vernier et al. (2015). Dots indicate the Angström exponent for the fire (red) and ATAL (green).

is in good agreement with the tropical upwelling velocity in current reanalyses (e.g. Abalos et al. (2015), Fig. 6). Similar ascending features are visible around the globe 0-25°N (Fig. C1). The reversed vertical transport of the aerosol particles in Fig. 1B compared to 1D (i.e. the observed descent in the northern latitudes and ascent in the tropics) reflects the contribution of the ascending and descending branch of the BDC. The average signal for the magenta box remains also until April 2018 at ∼19

km altitude. Based on the good agreement of the slope in the aerosol signal with downwelling and upwelling velocities by the BDC we hypothesize that the BDC played a role in both extratropical downward and tropical upward transport of the aerosol. To further verify the link between the enhanced aerosol extinction values in the Asian monsoon region (especially south of the center, as seen in Fig. 1) and the Canadian wild fire plume, individual SAGE III observations are further analyzed in Fig. 5, B2 to B3, to show that the enhanced aerosol extinction values (Fig. 1) can be assigned to the Canadian fire plume.

As a solar occultation instrument, SAGE III provides an enhanced sensitivity of the aerosol extinction signal. In order to provide better statistics for a daily mean and investigate a potential co-occurrence of the fire plume and the ATAL in this area, the denser data set in terms of space and time of the OMPS-LP is inspected in the black box. In Fig. 1E the ATAL is clearly visible for all three years displayed (2016-2018) in July and August up to 18 km altitude. The fire plume appears at altitudes between 16 and 21 km at the beginning of September 2017 and is not visible anymore at the end of March 2018. Both signals

appear as two distinguishable events on different altitude ranges (fire plume: 16-22 km, ATAL: <18 km). In Fig. 1E, the ATAL is mostly apparent in the troposphere (at altitude levels exhibiting a generally higher AMA confinement but as expected also with some distinguishable signal in the lowermost stratosphere), while the fire plume appears clearly in the stratosphere (at

altitude levels with generally weaker confinement), which indicates a clear separation between the two aerosol layers. During early September (earliest fire signal in Fig. 1E), the ATAL is already in the process of slowly declining.

The green solid line in Fig. 2A shows the average SAGE III profile for the black box from Fig. 1A, prior to any fire influence, i.e. $10^{th}$-$15^{th}$ August. It shows a clear ATAL signal with a broad peak located at ∼15 km altitude. This peculiar peak becomes even more evident in comparison with the average aerosol extinction signal during the same period outside the Asian monsoon box (black dashed line). The ATAL signal doubles the background aerosol extinction at 15 km altitude. The doubling of background conditions and the altitude range agree well with the ATAL signal found by Vernier et al. (2015) in the SAGE II and CALIOP

data set. There are no SAGE III measurements from July to August $10^{th}$ in the wider Asian monsoon area (green box from Fig. 1A). Considering the data coverage of OMPS and SAGEIII, the ATAL height of the two data sets are in a reasonable agreement (up to 18 km for OMPS and peaking at 16 km for SAGEIII). Looking at the green box between September $1^{st}$ and $5^{th}$ (red line) shows that the aerosol extinction of the fire plume reaching the Asian monsoon area is by a factor of ∼9 higher than the ATAL signal observed in August and by a factor of ∼18 higher than background aerosol conditions. The average peak altitude

of the visible fire plume is ∼18 km.

    From the average fire plume and ATAL spectral aerosol extinction profiles identified in Fig. 2A, spectral aerosol optical depth (AOD) are derived and shown in Fig. 2B. In Vernier et al. (2015), the time-evolution of the averaged ATAL AOD in July-August, between 1995 and 2013, is derived at 525 nm with a combination of SAGE II and CALIOP data, using different hypotheses of the LiDAR ratio value for the target aerosol layer. For this reason, the derived AODs are very uncertain. The

average July-August ATAL AOD estimated by Vernier et al. (2015) lies between 0.0030 and 0.0070. In our case, the average ATAL AOD at 525 nm is ∼0.0025. This is consistent with Vernier et al. (2015), considering that our value is averaged on a late monsoon period (i.e. August $10^{th}$ to $15^{th}$), when the dynamical barrier of the AMA starts to decline and, consequently, the ATAL concentration and AOD are possibly weaker than e.g. in July. The ATAL AOD decreases steeply with wavelength to values smaller than 0.001 at 1020 nm. The fire plume has significantly higher AOD values, i.e. from values higher than 0.010

at 380-520 nm to ∼0.0045 at 1020 nm. As an indicator for the average particle size in the aerosol populations the average Angström exponent is estimated for both ATAL and fire plume (Fig. 2B). It is calculated starting from the AOD values at 869 and 521 nm. A higher Angström exponent can be associated with the prevalence of smaller particles, while a lower exponent points at larger particles, on average. Angström exponents between ∼0.8 and 1.5 may be linked to aged biomass burning plumes (e.g. Müller et al. (2007) and Pereira et al. (2014)). On the contrary, higher Angström exponents (1.8 or greater) may point at

freshly nucleated secondary aerosol and/or younger biomass burning plumes (e.g. Müller et al. (2007)). The average Angström exponent has values of ∼1.9 for the ATAL and 0.9 for the fire plume (Fig. 2B). This is consistent with our identification of an aged fire plume, transported into a region previously dominated by a layer of freshly nucleated, smaller particles.

    To investigate the dynamics of the fire plume transport to the AMA region, an air mass origin tracer has been initialized between August $12^{th}$ and $14^{th}$ 2017 in the box over western Canada (green box in Fig. 3), using the CLaMS model. The point

in time and space of the initialization box was chosen according to the position and time of high observed IASI CO values due to the fire, which is shown in Fig. 4. The simulation with box initialization as presented here, provides an illustration of the possible large-scale transport pathways, but should not be taken for quantitative estimations for two reasons: $1^{st}$ The significant

self-rising feature of this plume (as observed in Khaykin et al. (2018) and Yu et al. (2019)) is not considered in this or any previous Lagrangian transport model and $2^{nd}$ as we chose a box shaped area to represent the start of the trajectories (green

box in Fig. 3A-D), it becomes evident that some air masses within the box do not belong to the fire plume. The model fire tracer was injected in the respective box throughout the layer 345-465 K. This approach was found to be very robust in terms of initialization levels, by initializing air masses on different potential temperature levels and on each day between August $12^{th}$ and $14^{th}$. Therefore, uncertainties arising from the observed time and injection altitude do not interfer with our line of arguments, and the fact that clear transport pathways emerge from the large initialization region further corroborates of our

5 results. After initialization, the tracer has been advected passively during the following weeks. This approach is similar to the one presented in Vogel et al. (2015): the plume is first transported eastwards, at latitudes >40°N and passes over Europe in early/mid-August (Fig. 3A). After reaching the Asian monsoon area at the end of August, a fraction of the fire tracer is partly transported along the eastern flank of the AMA circulation from the extratropics into the tropics (Fig. 3B). In the simulations, part of the plume even reaches the southern hemisphere (Fig. 3C). It is shown that the plume reaches the tropics (<10°N)

10 first through the AMA circulation (Fig. 3C). This is consistent with the SAGE III observations shown before. With the slow breakdown of the AMA, plume air masses mix into the area that has before been confined by the AMA transport barrier from the northern side (Fig. 3D). By mid-September most of the NH is filled with the artificial fire tracer at 380 K potential temperature (Fig. 3D). Similar features are seen at 400 K potential temperature (Fig. D1). This pathway of the fire plume transport to the tropics within the eastern flank of the AMA circulation is further confirmed by OMPS aerosol extinction observations (see Figure D2 of the supporting material).

A measurement profile with readily identifiable and vertically separated AMA and fire plume signatures is shown in Fig.

5A. Around 370 K, the profile inside the AMA shows no clear evidence of enhanced aerosol from the fire plume (Fig. 5A at ∼16 km altitude). This is consistent with the existence of the generally strongest confinement (transport barrier) at around 380 K (Ploeger et al., 2015). The generally weaker confinement at around 400 K compared to 380 K is reflected by the CO gradient and Montgomery stream function shown in Fig. 5B. The enhanced CO mixing ratios displayed in Figure 5B, indicate the entrainment of tropospheric tracers insider the AMA (e.g. Santee et al. (2016); Park et al. (2008)). The profile in Fig. 5 is

selected here, because of its location within the eastern flank of the AMA circulation (Fig. 5B), within the canonical north-south transport pathway from the extra-tropics to the tropics. Back-trajectories show that air masses from the altitude levels of the fire plume and ATAL peaks pass over partly different regions 9 days prior to the SAGE III measurement profile (Fig. 5C). For this study ultra-violet (UV) aerosol index measurements by OMPS and the position of detected enhanced aerosol extinction values by CALIPSO are displayed in Fig. 5C. While CO mixing ratios are a fire indicator for 'fresh' plumes, enhanced aerosol

can be traced over longer time scales. Because of the spatial distance between the fire plume origin and the Asian monsoon region, aerosol extinction values rather than CO measurements are taken as an indication for the fire plume. About 40% of the red back-trajectories (initialized at the fire plume peak altitude) pass over areas with fire influence. Air masses measured at 17-18 km altitude in Fig. 5A can be understood as a mixture of air masses, partly influenced by the fire and partly coming from other regions; in any case, the fire plume particles signature on the aerosol extinction is predominant. Air masses measured at 15-16 km reasonably follow the AMA circulation and pass through the polluted and convective South-East Asian and Indian

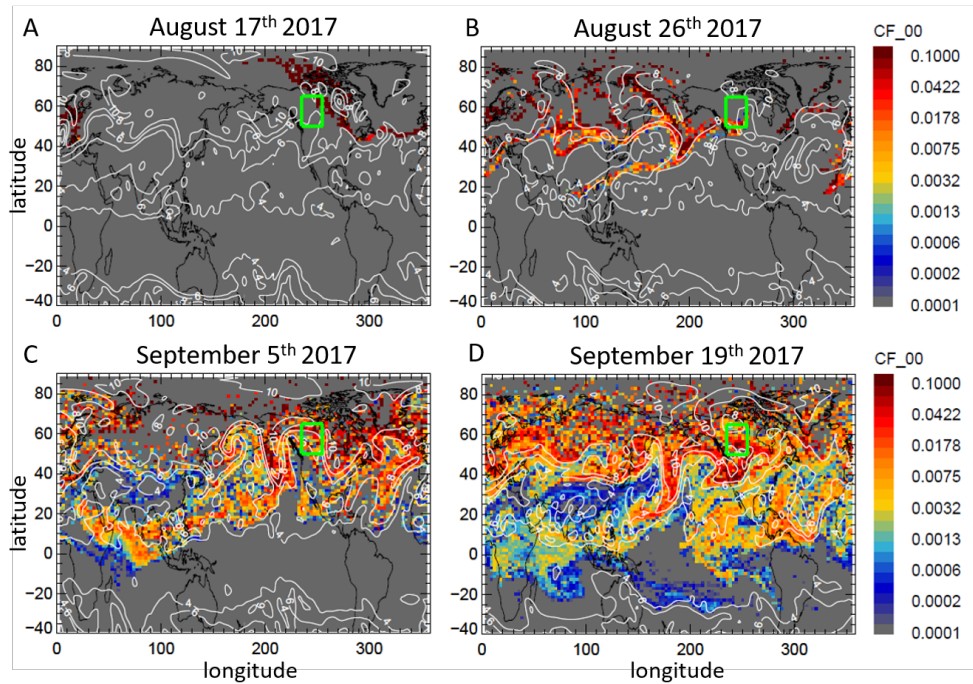

**Figure 3.** Selected maps at 380 K potential temperature on (A) Aug $17^{th}$, (B) Aug $26^{th}$, (C) Sep $5^{th}$ and (D) Sep $19^{th}$ at 380 K for the CLaMS simulation for the fire tracer initialized in the green box (area: 50<lat<65, 235<lon<255, potential temperature: 345 K<theta<465 K) between 345 K-465 K potential temperature. CF_00 represents the fraction of the air mass originating from the green box. The position and time (Aug $12^{th}$ to $14^{th}$ 2017) of the initialization of the fire tracer has been chosen according to the first appearance of enhanced CO mixing ratios in the stratosphere above western Canada, observed by IASI (as shown in Figure 4). White lines represent PV contours. This approach is similar to the one presented in Vogel et al. (2015).

region. Therefore, the two peaks in Fig. 5A, one at 17-18 km altitude (∼395K) and one at 15-16 km altitude (∼370 K), can be associated with the fire plume and with the ATAL respectively. With the limiting spatial resolution of remote sensing instruments, it cannot be concluded whether both air masses have mixed. More examples are discussed in the supplements: one profile showing enhanced aerosols due to the fire, originating from the plume (Fig. B2), one profile showing both enhanced aerosols from the plume and ATAL (Fig. B1) and one profile with a visible ATAL signal and no fire signature, due to the dynamical position of the measurement (Fig. B3).

## 4 Regional climate impact

The radiative properties of the fire plume, once in the AMA region, are studied and compared with those of the observed late ATAL layer. Starting from the spectral aerosol extinction from the SAGE III observations for the mean fire plume and ATAL of Fig. 2A, the radiative impacts have been estimated by means of the shortwave (300.0 to 3000.0 nm) daily average clear-sky

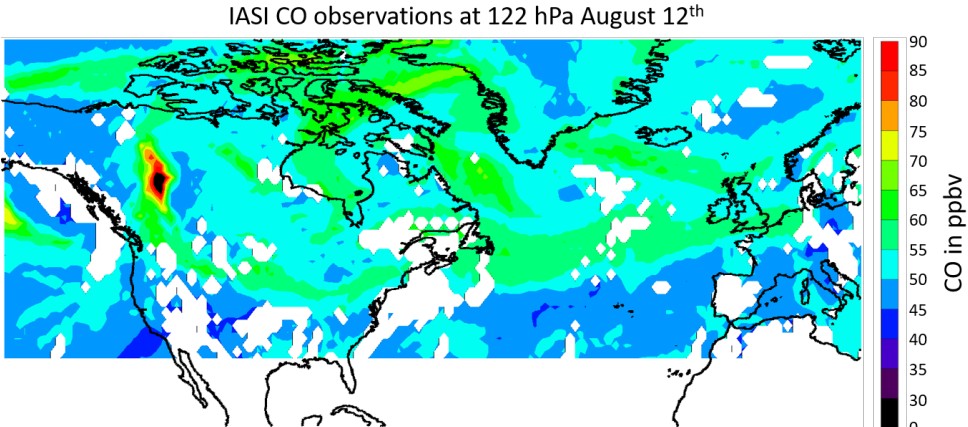

**Figure 4.** IASI-SOFRID CO mixing ratios at 122 hPa for the morning (9:30 Local Solar Time) of the $12^{th}$ of August. The high CO plume detected over Canada is used to determine the initialization box chosen for the CLaMS simulation in Fig. 3.

direct radiative forcing (RF) at the surface and at the TOA, as well as their f ratio (Fig. 6). The f ratio is defined as the ratio between surface and TOA RF (e.g. Di Biagio et al. (2010) and Sellitto et al. (2016)). Even though the aerosol extinction data used in our simulation have been observed, hypotheses are necessary for the single scattering albedo (SSA) of the two aerosol layers. To cope with the limited knowledge of this parameter, we run the radiative transfer simulations multiple times using different realistic SSA values: from 0.90 to 0.93 for the advected fire plume and from 0.97 to 0.99 for the ATAL. Values higher

than 0.90 for aged fire plumes are found by e.g. Haywood et al. (2003). This points at the presence of less absorbing features with respect to fresh biomass burning soot because of the progressive coating of condensed sulfates and/or organics (Ditas et al. (2018) and references therein). In addition, SSA for boreal forests fires have, on average, a higher SSA than tropical forests fires (Wong and Li, 2002). The optical properties of this fire plume have been observed with a ground-based LiDAR, on 22 August in Europe, by Haarig et al. (2018). They report a SSA of 0.80 in the visible spectral range, which is typical of pure-soot

particles. Nevertheless, our radiative simulations are representative of a plume at least 2 weeks older than the one sampled by Haarig et al. (2018) and with quite likely less absorbing (in terms of absorption to scattering ratio) sulphate/organics-covered soot particles. Ditas et al. (2018) have shown that SSA, for a biomass burning aerosol plume, is strongly dependent on the coating thickness of core black carbon particles. For aged fire plumes, a particle-to-core ratio of 4 or bigger was observed with in-situ aerosol observations on aircraft platforms (Fig. S12a of Ditas et al. (2018)). In these cases, the particles SSA has

values of 0.90 or bigger (Fig. S12b of Ditas et al. (2018)). Therefore, we select 0.90 to 0.93 as the interval of SSA for the particular aged fire plume investigated in our paper. The ATAL is considered as principally composed of freshly nucleated secondary organic and sulfate aerosols, both characterized by very high (near-1) SSA. Nevertheless, the precise composition of the ATAL and its variability are still largely unknown. Then, starting from the idea of very reflective aerosols, we have extended the interval of SSA for the ATAL, to smaller values (i.e. down to 0.97). To the best of our knowledge, ours is the first

estimation of the surface direct RF and f ratio of the ATAL. Our estimation of the TOA RF for the ATAL (about -0.05 W/m$^2$)

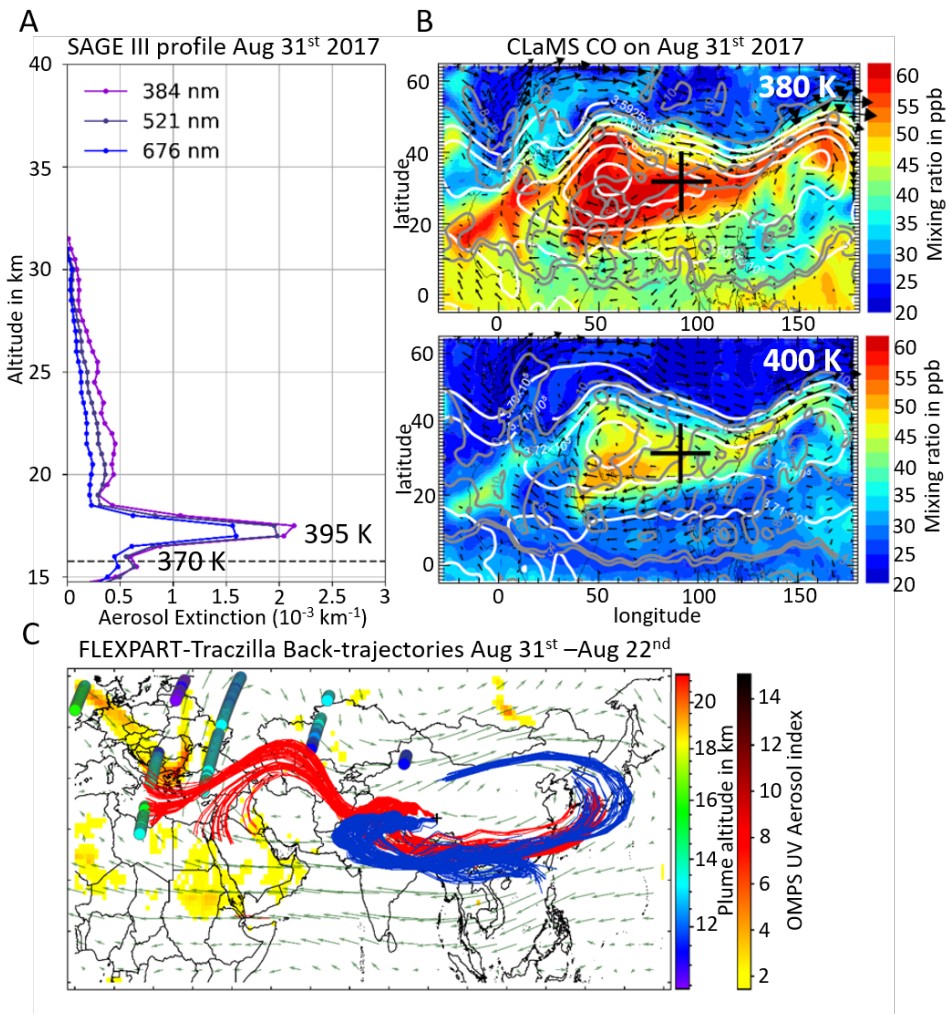

**Figure 5.** (A) SAGE III aerosol extinction values for the profile taken on August $31^{st}$ 2017 at 31.53°N and 91.56°E. The dotted black line represents the given tropopause height by the SAGE III data set. (B) CLaMS CO map at 380 and 400 K on August $31^{st}$ 2017. Gray lines represent the PV contours at 3.6 PVU. The determination of the AMA PV transport barrier works only on a few days in 2017. White lines show the Montgomery stream function. The black cross represents the profile position. (C) OMPS UV aerosol index (yellow-reddish areas) and the plume altitude based on CALIOP observations (blue to green dots), which show enhanced aerosol values due to the fire plume on August $22^{nd}$ (9 days prior to the SAGE III measurement profile in A). Superimposed: Every $5^{th}$ of the 1000 9-day back trajectories calculated using TRACZILLA starting from the respective altitude and position of the two peaks shown in A. The blue and red trajectories correspond to the peaks at 15-16 km (370 K) and 17-18 km (395 K) respectively in A.

is significantly smaller than the one from Vernier et al. (2015) (-0.12 W/m$^2$, also shown in Fig. 6). This might be related to the fact that our estimation has been made using observed ATAL aerosol extinction profiles for a late monsoon period (August).

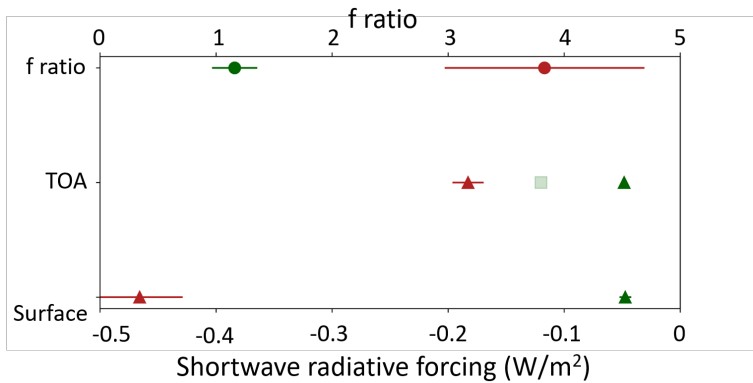

**Figure 6.** Daily average clear sky top of the atmosphere (TOA) and surface direct radiative forcing (triangles), and f ratio (dots), for the mean fire plume and ATAL profiles of Fig. 2A. Green represents the calculations for the ATAL, red for the fire. The variability of the SSA in our calculations is reflected by the error bars. The clear sky TOA radiative forcing estimation for the July-August ATAL of Vernier et al. (2015) is also shown (semi-transparent green square).

The estimated shortwave surface RF is about -0.05 W/m$^2$, consequently, the ATAL f ratio is $\sim$1. This value is typical for very reflective aerosol layers, as expected for a layer mostly composed of condensed sulfates or organic aerosols. On the contrary, the estimated shortwave RF for the advected aged fire plume is about -0.18 W/m$^2$ at TOA and about -0.46 W/m$^2$ at the surface, leading to an average f ratio of $\sim$3.8, typical for significantly absorbing layers (e.g. Sellitto and Briole (2015)). From the TOA
5   RF calculations for the fire plume and ATAL, it can be concluded that the regional climate impact of the fire plume is up to 4 times (late ATAL, our estimation) and 2 times (peak ATAL, estimation by Vernier et al. (2015)) larger than the one of the ATAL. Our RF estimation for the fire plume is consistent with the estimated RF for biomass burning from wildfires of Ditas et al. (2018). The fire plume TOA RF estimated here in the tropical UTLS has the same order of magnitude as a moderate volcanic eruption. For example, Haywood et al. (2003) have estimated the mean RF of the 2010 Sarychev aerosols to about
10   -0.25 W/m$^2$ once dispersed in the NH stratosphere. While here we do not attempt to estimate the impact of the aged Canadian fire plume over the whole NH, we provide an estimation of an already widespread fire plume (see Fig. 3C and D). In addition, the calculated values of the f ratio and TOA RF suggest that while the fire plume has this significant cooling impact on the climate system, it can produce an additional local heating in the layer where it is located, for f ratio values considerably larger than 1 (Sellitto and Briole, 2015).

15   **5   Conclusions**

The circulation of the AMA has effectively transported Canadian fire plume air from northern latitudes ($\sim$40°N) around its eastern flank into the tropics, where the air has further been lifted with the ascending branch of the BDC.
Although the highest extent of the fire plume in the stratosphere is seen in higher latitudes (>40°N), a plume signal appears

at the end of August 2017 in the lower stratosphere (17 km) above the AMA/layer of confinement. The signal remains visible in the Asian area until April 2018. The diluted fire plume exceeds the concentrated ATAL signal, on average, in the Asian monsoon region by a factor of 9 in terms of measured aerosol extinction (September $1^{st}$-$5^{th}$ compared to August 10-15$^{th}$ 2017). Within the AMA region in 2017 (25-38°N and 40-95°E) and also south of the AMA (5°N-25°N and 40°E-95°E) an enhanced aerosol extinction signal is visible above the tropopause in the second half of August. This signal remains at ∼19 km until April 2018. We found no evidence that the fire plume passes the barrier, mixing with the air masses inside the AMA. We conclude that the fire plume has largely bypassed the AMA, as observed in the past for the Sarychev plume.

Even though recent moderate volcanic eruptions have had bigger impacts on the aerosol budget in the global stratosphere, we show that also extreme fire events, like the Canadian fires in 2017, can have an impact on the global atmosphere by transport of the plume into the tropics via the eastern flank of the AMA circulation. These fire events may even occur at high northern latitudes and are still subject to efficient transport to the tropics. For this event, we estimate a significant regional climate impact, exceeding the ATAL climate forcing by a factor of 2 to 4, and our calculations suggest a significant cooling of the NH climate system, comparable to a moderate volcanic eruption. The partially absorbing nature of aged fire aerosols may lead to a significant local heating of the layer where they reside. While in this study we present one case example of the plume that originated from the Canadian wildfires in 2017, it has to be noted that extreme fire events like this are expected to occur more frequently due to climate change (Field et al., 2014). Already in 2018 another series of record breaking wildfires has been reported (Lindsay, 2018). For future extreme wildfire events the AMA circulation will provide a pathway for long-range transport of atmospheric pollutants and fire plumes each year from the NH to the tropics and possibly to the southern hemisphere and the stratosphere.

Most of the Canadian fire aerosols injected into the northern hemisphere stratosphere have been transported to the north, descending back to the troposphere via the lower branch of the BDC (Yu et al., 2019). In this study, we analyze one specific southward transport path way: the transport in the lower stratosphere within the easterly jet together with the north-south anticyclonic flow at the eastern flank of the AMA. We show that this is an efficient transport pathway from northern latitudes to the tropics. However, other dynamical processes bringing air masses from the mid-latitudes to the tropics are possible and not investigated in this study: for instance, the easterly jet in the lower stratosphere does not appear only at one specific latitude, but can also allow for a southward (also northward) propagation of air masses. Furthermore, fire plume air masses injected in the mid-latitude upper troposphere to the tropics with subsequent uplift to the stratosphere could be considered. Beside the AMA, which is the by far largest periodically reoccurring anticyclonic flow system in the lower stratosphere on Earth, dynamical effects of other monsoon systems have the potential of transporting air masses southwards (e.g. the North-American monsoon and the West-African monsoon) in the lower stratosphere.

*Data availability.* The aerosol extinction data sets from SAGE III-ISS v5.1 are available at https://eosweb.larc.nasa.gov and OMPS v1.5 at https://daac.gsfc.nasa.gov/. The model and simulation data may be requested from the corresponding author: the CLaMS model data

(f.ploeger@fz-juelich.de), the UVSPEC input and output files for the radiative forcing calculations (pasquale.sellitto@lisa.u-pec.fr) and the TRACZILLA back-trajectories (silvia.bucci@lmd.ens.fr).

*Author contributions.*  C.K., G.B. P.S. designed the research, analyzed and interpreted the data. F.P., S.B., P.S provided simulations and calculations, B.L. L.W.T., G.T., B.B., E.L.,S.K. provided data and data analysis. M.H.,A.B., F.J.,N.B. were involved in discussion and data interpretation. All co-authors contributed in writing the paper.

*Acknowledgements.*  The authors C.K., G.B., P.S., S.B., F.J., A.B. and B.L. acknowledge the support of Agence Nationale de La Recherche under grant ANR-17-CE01-0015 (TTL-Xing). This work was undertaken as part of WP7 of the VOLTAIRE LabEx (VOLatils – Terre, Atmo-
sphère et Interactions – Ressources et Environnement), convention number ANR–10–LABX–100–01. Furthermore, the authors acknowledge CNES for some financial support of IASI activity at Laboratoire d'Aérologie in the framework of the TOSCA-IASI project. The providers of the LibRadtran suite are gratefully acknowledged. C.K. was partly funded by the Deutsche Forschungsgemeinschaft (DFG, German Research Foundation) – 409585735. F.P. was funded by the Helmholtz Association under grant VH-NG-1128 (Helmholtz Young Investigators Group A-SPECi). S.B. was supported by the EU FP7 grant 603557, CEFIPRA5607-1. SAGE III/ISS data were obtained from the NASA
Langley Research Center Atmospheric Science Data Center (10.5067/ISS/SAGEIII/SOLAR_BINARY_L2-V5.1). Furthermore, the authors acknowledge the National Aeronautics and Space Administration (NASA), the SAGE III-ISS and OMPS teams. The authors thank Jean-Paul Vernier for helpful discussions and David Huber for technical support.

## Appendix A:  Break down of the AMA circulation criterion

The criterion for the breakdown of the AMA circulation in 2017 is chosen according to the strength of the easterly jet that forms the southern branch of the anticyclone and displayed in Fig. A1. The jet has its maximum at 12 km altitude (196 hPa). Hence, the criterion is based on this altitude. For the years 2010-2017 the jet intensifies on average from April to July, with a
highest intensity in July (green curve). While the jet intensifies, it moves northwards to ∼5°N. Especially in 2017 (red curve), the data show a much faster onset than decline, which is not the case for an average year between 2010 and 2017 (black curve). From those data it can be concluded that the anticyclone is still there until the end of September 2017.

## Appendix B:  Supporting material for Figure 1 and 5

Another SAGE III aerosol extinction profile with two peaks, one at 15 km (tropopause height, corresponding to a potential
temperature of 365 K) and one at 17 km (corresponding to a potential temperature of 380 K) is shown in Fig. B1. The magnitude of the aerosol extinction peak at 15 km agrees well with the ATAL signal observed in Fig. 2. To put the observed measurement profile of Fig. B1A in the context of the dynamical/meteorological situation on the day of the SAGE III measurement profile, the strength and position of the AMA have been investigated (Fig. B1B) and back trajectories calculated. While the 9 day back trajectories from the peak at 17 km altitude (red trajectories), reach out to western Asia, the blue trajectories, corresponding to

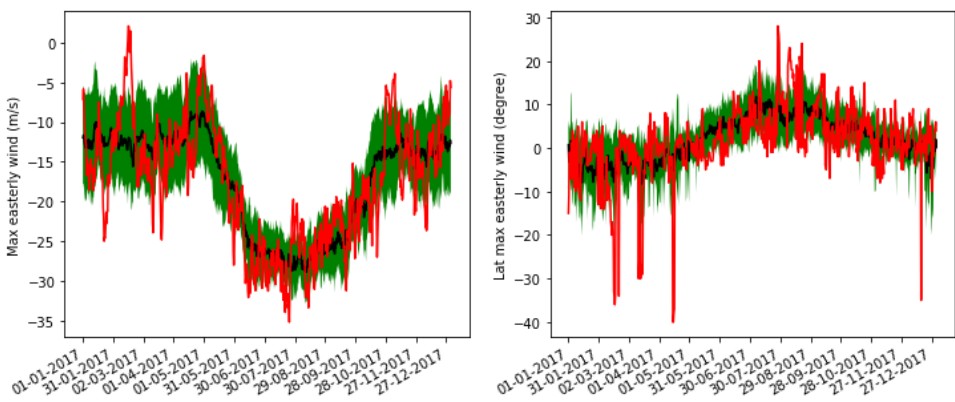

**Figure A1.** Maximum easterly wind (m/s) and latitude of the maximim wind (in degree) produced from ERA 5 (Copernicus Climate Change Service (C3S), 2017). Climatology from twice-daily data 2010-2017 in black, with +/- one standard deviation in green. Respective data for the year 2017 in red.

the lower peak at 15 km do not reach as far to the west. At this time the very first fire plume signal is observed in western Asia, then transported within the next week over central and east Asia ((Khaykin et al., 2018) in Fig. 3). Consequently, the higher peak at 17 km altitude (with air mass influence from far western Asia on the $21^{st}$ of August) may be interpreted as a mixture of air masses partly originating from the Canadian fire plume. The lower peak at 15 km altitude (with 9- day backtrajectories reaching back to regions where the fire plume has not been transported to yet and staying closer to the center of the AMA) can be associated with the ATAL signal (Fig. B1). Fig. B2A shows a clear aerosol extinction enhancement at 18 km altitude, equivalent to 445 K potential temperature. At the altitude of the aerosol extinction peak, the calculated Angstrom exponent is ∼1.2, suggesting that the peak indeed results from the aged fire plume. Fig. B2B shows the position of the SAGE III profile within its dynamical surrounding. The CLaMS CO tracer in Fig. B2B gives an indication about the position of the AMA center with its enhanced tropospheric trace gases and does not show 'real' CO abundances (e.g. the model CO mixing ratios depend strongly on the used boundary condition from AIRS observations and on the representation of tropospheric transport processes, see Pommrich et al. (2014)). Even though the profile has been taken in early September, a time when the transport barrier is generally in the phase of slowly dissolving, enhanced artificial CO with isolated features is still visible at 400 K, which confirms the long lifetime of the AMA particularly in 2017 shown in Fig. A1. A group of perturbed back-trajectories (1000), obtained using the TRACZILLA Lagrangian model, has been launched from the center of the peak seen in Fig B2, determined form the SAGE III observation. All 1000 calculated back trajectories pass over northern Canada ∼18 days prior to the SAGE III measurement profile at a pressure level of 85.87 ±4.37 hPa (Fig. B2C). IASI CO observations show highly increased CO values at a 85 hPa pressure level. It is easily identified as due to the injection of the Canadian fire plume into the stratosphere 18 days prior to the SAGE III measurement profile (respective to the end points of the trajectories). Consistent with Fig. B2C, the upper aerosol extinction peak in Fig. B2A can convincingly be associated with the Canadian fire plume.

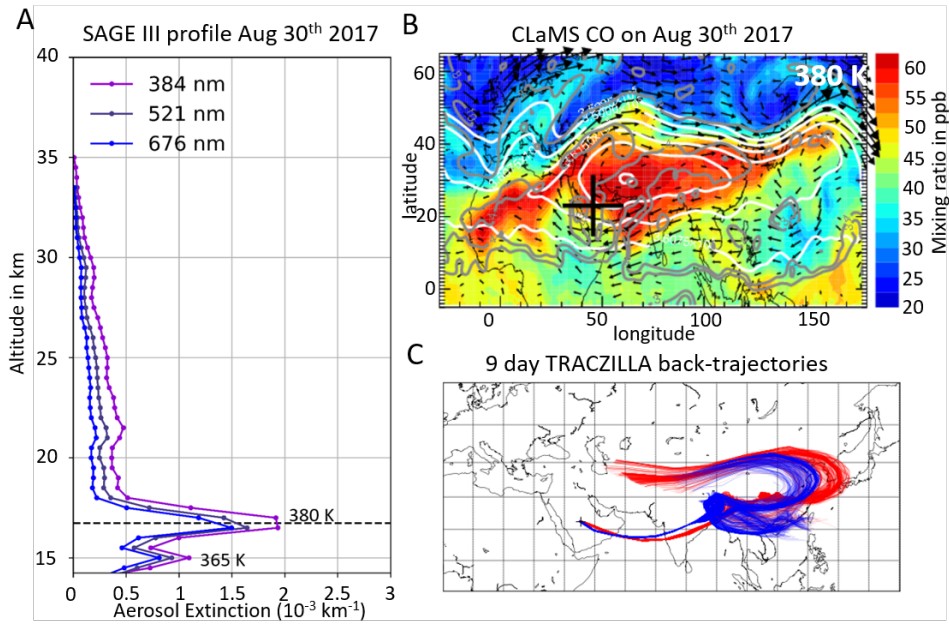

**Figure B1.** (A) SAGE III aerosol extinction values for the profile taken on August $30^{th}$ 2017 at 22.47°N and 45.18°E. The dotted black line represents the given tropopause height by the SAGE III data set. (B) ClaMS CO-map at 380 K. Grey lines represent the PV contours at 3.6 PVU. The determination of the AMA PV transport barrier works only on a few days in 2017. White lines show the Montgomery stream function. The position of the measurement profile from A is indicated with the black cross. (C) The 1000 9-day back trajectories respective to altitude and position of the two peaks shown in A. The beginning of the blue trajectories corresponds to the peak at 15 km altitude (365 K), the red trajectories to the peak at 17 km altitude (380 K) in A.

Pure ATAL signals without any fire signature often appear in the SAGE III data set at 15 km altitude or slightly above, with Angstrom exponent mainly between 2.0 and 2.5. At the peak altitude 16 – 17 km we calculate an Angstrom exponent of 2.2. Therefore, the enhanced aerosol extinction values in Fig. B3A can be associated with the ATAL signal, even though it is 1-2 km higher than other ATAL signals seen with the SAGE III data set (which could be a sign of an ATAL air mass leaving the dissipating AMA 'bubble' (Yu et al., 2017)). On the northern side of the AMA a high PV gradient indicates a strong transport barrier (B), related to the strong westerly winds of the subtropical jet. Therefore, the profile taken at the border of the AMA is dynamically clearly in the center of the AMA and it is not surprising that no fire plume signal is visible Fig. B3A. Fig. B3C shows the 9-day back trajectories from the ATAL signal in Fig. B3A. All 1000 trajectories only reach areas of the core anticyclone shielded by the transport barrier, following the wind fields shown in Fig. B3B.

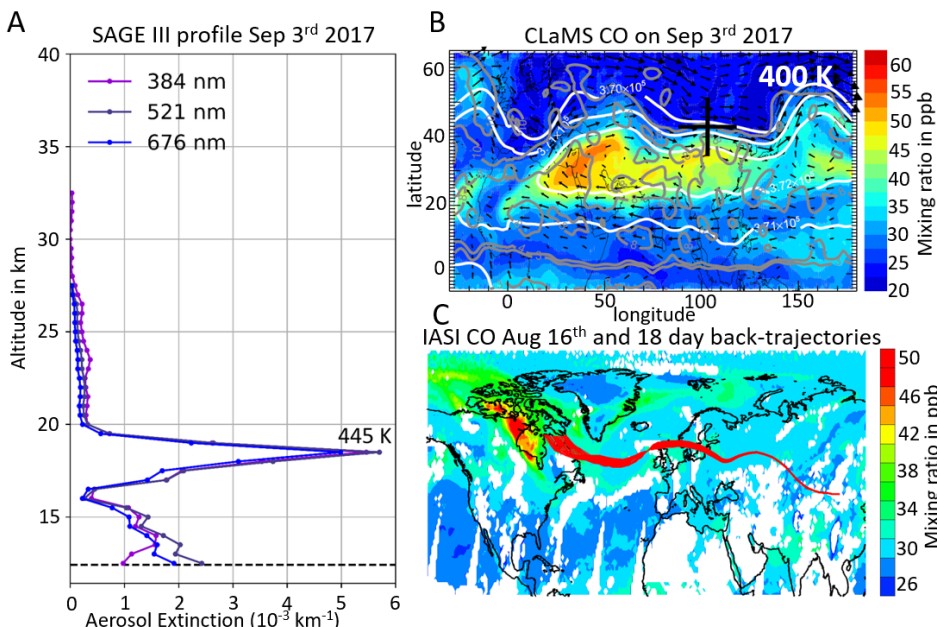

**Figure B2.** Same as for Fig. 5 and B1 for September $3^{rd}$ 2017 at 42.94°N and 103.8°E. The dotted black line represents the given tropopause height by the SAGE III data set. (B) CLaMS CO map at 400 K on the $3^{rd}$ of September 2017. (C) IASI CO map, integrated over 12 hours at 85.18 hPa 18 days before the SAGE III profile measurement in A. Superimposed: Every $5^{th}$ of the 1000 18-day TRACZILLA kinematic back trajectories initialized at the geographical position and altitude of the fire plume peak in A.

## 5 Appendix C: Supporting material for Figure 1D

As the upwelling of the BDC in the tropics is a feature usually displayed on a bigger scale, a larger area covering all longitudes in the northern hemispheric tropics (0-25°N) was chosen (Fig. C1). The ascending feature of enhanced aerosol extinction values (as seen in Fig. 1D) are clearly visible in Fig. C1 and can therefore be attributed to the rising branch in the tropics of the BDC.

## 10 Appendix D: Supporting material for Figure 3

Once injected to the stratosphere the initialized fire tracer follows very similar patterns at different potential temperature levels as seen when comparing Fig. 3 with Fig D1. Fig. D2 shows OMPS aerosol extinction measurements, supporting the transport pathway simulated by ClaMS in Fig. 3. The fire plume signal approaches the Asian region from the west and is transported into the tropics via the eastern flank of the AMA. To exclude a biased signal due to clouds, the altitude 18.5 km has been chosen.

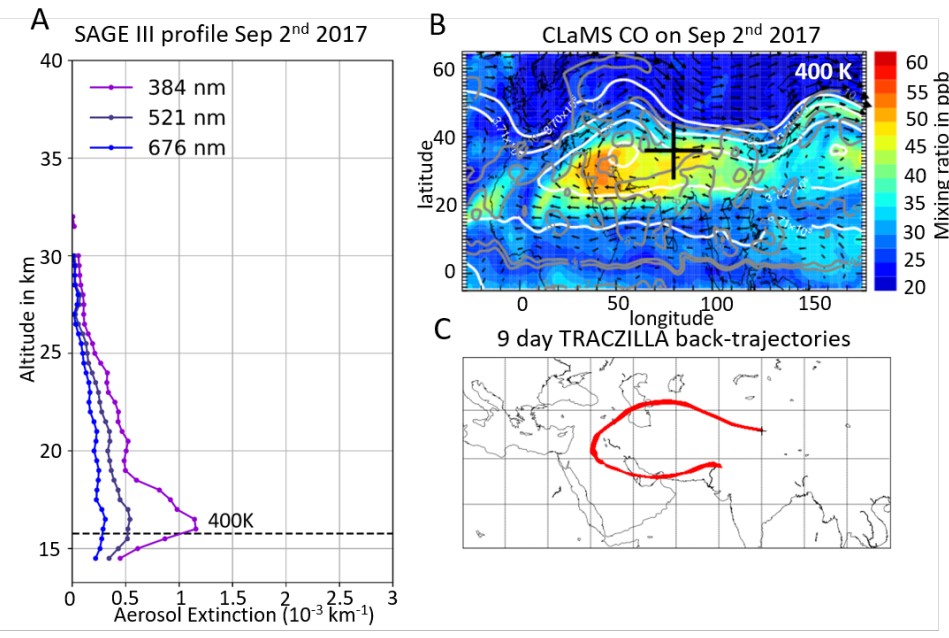

**Figure B3.** ATAL signal in the Asian region without fire influence: (A), (B) as in Fig. 5,B1 and B1 for the profile measured on the $2^{nd}$ of September 2017 at 35.96°N and 80.04°E. (C) 9-day TRACZILLA kinematic back trajectories from the peak position in A at 400K.

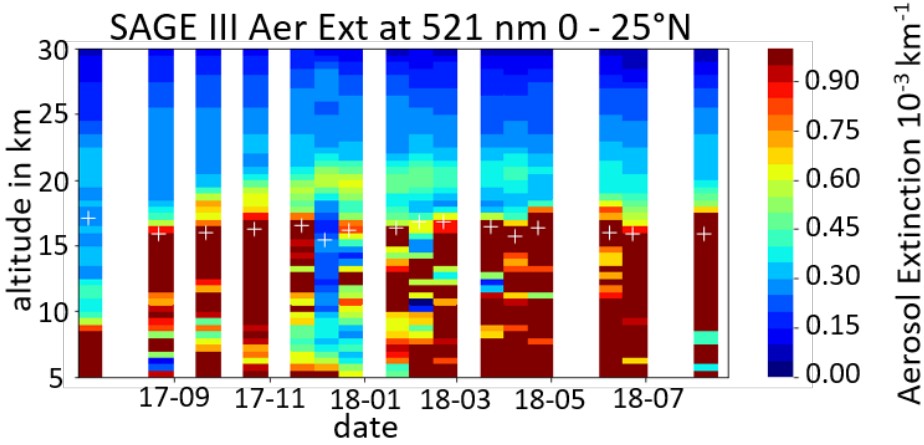

**Figure C1.** Same as for Fig 1D, but considering all measurements from 0-25°N.

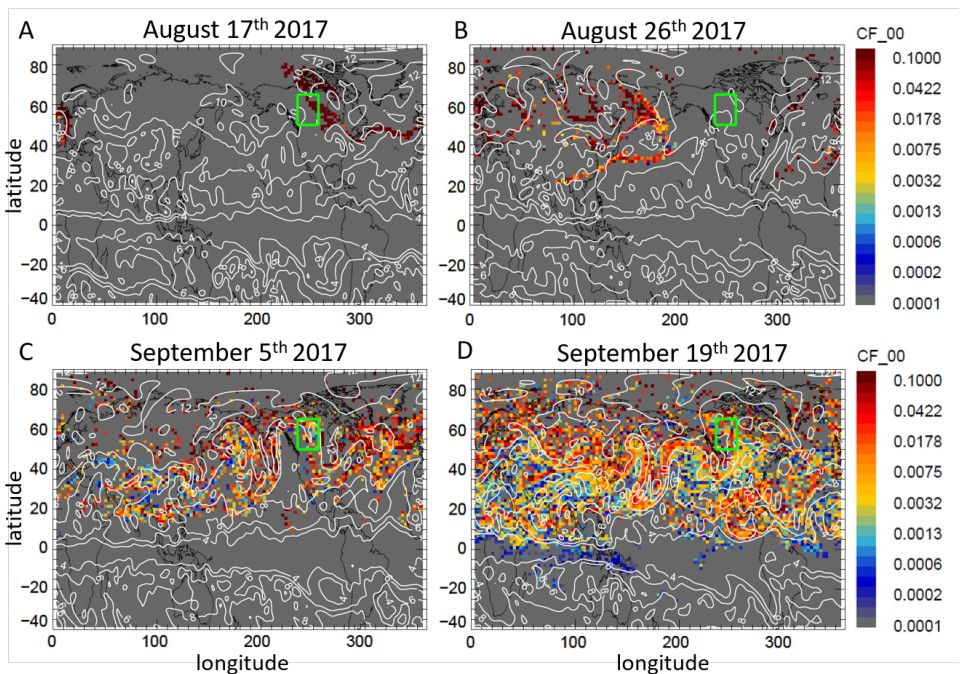

**Figure D1.** Same as Fig. 3 at 400 K potential temperature.

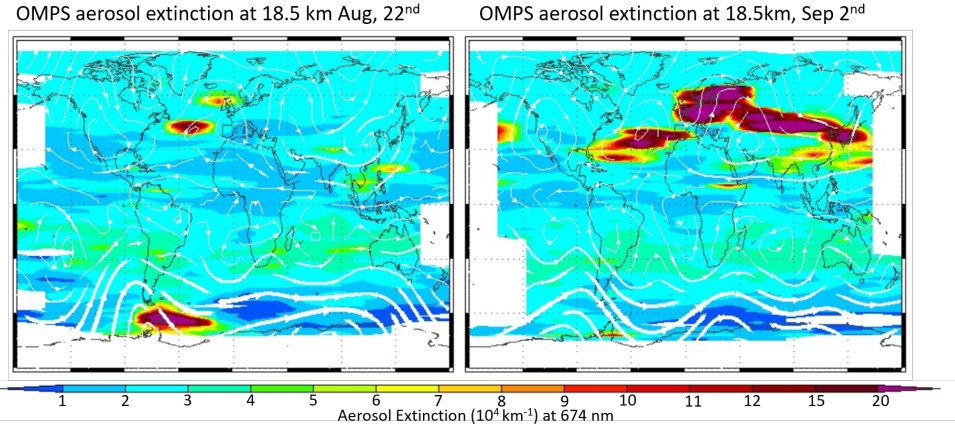

**Figure D2.** OMPS aerosol extinction observation 675 nm at 18.5 km altitude on August $22^{nd}$ and September $2^{nd}$ at 675nm. White lines are the MERRA winds.

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
