# Peer review of "Transport of the 2017 Canadian wildfire plume to the tropics via the Asian monsoon circulation"

_Atmospheric Chemistry and Physics, 2019_

## Referee Comment (RC1) · Anonymous Referee #2 · 9 May 2019

Kloss et al. (2019) reported the transport pathway of Canadian wildfire plume. This study suggests that Asian summer monsoon can effectively transport the plume to tropics mainly from the eastern side of the anticyclone airflow. Similar findings are reported in a previous paper (Wu et al., 2017, ACP). In Wu's paper, a volcanic plume is transported equatorward via the Asian summer monsoon. Effectively this is another case study on the transport pathway in the upper troposphere and lower stratosphere associated with Asian summer monsoon anticyclone. I find this study interesting and useful to communities because it reveal some details in the transport of the 2017 Canadian wildfire, an record-breaking extreme fire event that reached the stratosphere in recent decades. Some suggestions are attached.

[Figure]

Page 6 starting from Line 15: "...in the whole NH >40N mid to end of August..." I am not sure I understand Fig. 1B, is there any data shown from mid to end of August 2017? "The aerosol signature descends with 2 mm in altitude per second" Could you please draw a line on Fig.2B to show the decline slope? It is hard to tell by eyeball. "Hence, the descent of the aerosol is due to sedimentation" Well, I believe sedimentation plays an important role. However, what about the longitudinal dilution and cross-latitude transport from higher latitude to lower latitude (not necessarily happen in ASM region)? How do those affect the 5 km/month rate derived in the manuscript?

Fig.1E: From OMPS, ATAL is mostly in troposphere; while from SAGEIII (Fig.2A,4A), ATAL's peak extinction is above 15km. Please explain why OMPS' ATAL is lower.

Any stratospheric adjusting is taken into consideration in RF calculation especially for the 2017 fire plume with absorbing substance?

In terms of equatorward transport of the plume, any other mechanisms/pathway can happen? For example, can the plume be lifted higher in the stratosphere in mid-hight latitudes, and then been transported to the tropics? Is there any way you can quantify/compare the relative fraction of plume transported to tropics via the two ways respectively?

Fig.4 I am a little lost here: a. In 4A, which is ATAL and which is fire? b. Is CO in 4B and 4C associated with fire at all? From Fig.3, authors suggest that ASM barrier prevents fire smoke mixed in; If CO can be mixed in, why not aerosols?

Minor: Fig.3, why there are 2 identical color bars?

---

## Referee Comment (RC2) · Anonymous Referee #1 · 14 May 2019

The paper is well written and deals with observations and simulations of the unique, record-breaking northern hemispheric stratospheric smoke event that occurred in 2017-2018. Large amounts of biomass burning smoke (from bush fires in western Canada) reached the upper troposphere and (later on) the stratosphere up to about 20 km height and circled around the globe. The paper shows that the smoke even reached tropical regions via the Asian monsoon circulation.

The paper provides very interesting smoke observations (SAGE III, OMPS). As a suggestion, one could try to compare the observations with respective lidar observations (Hu et al., ACP, 2019, Haarig et al., ACP, 2018, CALIOP from September 2017 to March

2018).

Regarding the simulations: In Haarig et al. (2018), they show a size distribution (the soot showed a pronounced aged accumulation mode, but no coarse mode) and they found single scattering values of 0.74 (355nm), 0.8 (532nm), and 0.83 (1064nm). In your simulation you use SSA of 0.9-0.93. This is quite high for soot! Any comment? Maybe, another simulation with SSA of 0.8-0.85?

Please also check the paper of Ditas et al. (PNAS, 2018/19) concerning their simulation of the impact of soot on the radiation field.

---

## Short Comment (SC1) · 24 May 2019

I am compelled to offer some questions, comments and suggestions to help improve the research and paper of Kloss et al..

Abstract. In the first sentence ("...reached the tropics, and subsequently the tropical stratosphere...") Kloss et al. seem to suggest that the Canadian smoke plume, upon entry into the area of the AMA, had a discernible tropospheric component. Only subsequently was it lofted into the stratosphere by the BDC according to this claim. This is a fairly provocative claim. However I could not find any evidence given or figures showing upper tropospheric smoke adjacent to and wrapping around the AMA. They attribute all

the aerosols displayed below the tropopause to the ATAL. The evidence in Figure 1 and Khaykin et al. (2018) shows that by late August the smoke near the AMA was already at stratospheric heights and potential temperatures. If my understanding of the claim set forth in the abstract is correct, to defend it would require two things. 1. an unambiguous discernment of upper tropospheric smoke upstream of the tropical observations, and 2. evidence ruling out quasi-isentropic transport of the observed stratospheric smoke to the tropics. If on the contrary it is acknowledged that the smoke moving into Asia in late August was already spanning the lower stratosphere (as Khaykin et al. (2018) show) then it is hard to defend the abstract's claim convincingly.

On a technical but important note, the Abstract mentions "July" as part of the Canadian smoke event. There is no evidence here or in other papers that July was in play. This wording should be removed.

Introduction, L2-3. The manuscript stipulates that pyroCb activity is the source pathway for this plume. Hence it is critical to accurately establish the pyroconvective source. That is best done by citing Peterson et al. (2018) in this sentence. Peterson et al. give detailed and accurate constraints on both the pyroCb injection in the Pacific Northwest and the 3D footprint of the pyroCb plume on 14 August. Khaykin et al. (2018) points the reader to fires that did not exhibit pyroCb activity. (Sergey and I have had a personal communication on that matter.) Hence that paper is not fitting as a citation here.

On that topic, the choice of initializing CLaMS over three days centered on a box that is neither focused on the Pacific Northwest pyroCbs nor the pyroCb plume on a subsequent day seems destined to introduce many spurious or useless trajectories. The growing realization that there was significant diabatic lofting of the smoke further diminishes the applicability of the CLaMS construct and setup. Consequently little confidence can be gained from a set of these trajectories at a single potential temperature surface (especially since the plume was lower than 380 K in the first days (See Fig. 4 of Khaykin et al. (2018)).

Introduction, L29. Of the 3 papers cited on this line, only one postulates the Nabro troposphere-ASM -convection pathway: Bourassa et al. (2012). Fairlie et al. dispute that claim. Sellito et al. seem to be noncommittal on the pathway. Considering that Kloss et al. are apparently attempting to draw parallels with the Nabro publications and the 2017 AMA/smoke interaction (P2, L30), it is important to accurately portray the literature on the Nabro event.

P3, L22. Why was it decided to use "cloud unfiltered" SAGE 3 data? Thomason and Vernier (ACP, 2013) were compelled to go to great lengths to adopt a rigorous cloud clearing in SAGE II data for the study of tropospheric aerosols (indeed the ATAL). For inadequately constrained data sets such as SAGE and OMPS it is essential to either attempt aerosol-cloud discrimination or acknowledge that the tropospheric information content is uncertain. This is especially true for a regime like the particularly cloudy ASM.

P6, L17. Like one of the reviewers, I do not see evidence of descent. In fact it can be argued from this figure that aerosol is ascending. Indeed Khaykin et al. (2018) show that the extratropical smoke plume height increased dramatically, presumably due to diabatic forcing. What is the indicator of descent?

P6, L20. I don't see any difference in the extinction pattern after mid-April as compared to just prior to mid April. In fact tropospheric extinction appears to be saturated red throughout the timeline. I refer back to my comment above regarding cloud contamination and suggest that it is not possible to argue that the preponderance of the unfiltered tropospheric extinction signal on display is from aerosol.

P6, discussion of Fig. 1C. The value and information content of this figure panel is not obvious. As the authors state, detailed interpretation of smoke layers is hindered by the lack of filtering. In addition, half of the period rendered is the winter season, when there is no anticyclone and confinement. Presumably smoke aerosols would be in evidence in any other longitudinal sector in the winter. Hence some additional explanation of the

meaning of Figure 1C is called for.

P6, L29. Like the discussion of descent earlier, it is not evident what feature suggests ascent in Figure 1D. Moreover, there are additional plausible explanations for a sloping aerosol feature in a time series set in a localized domain. For instance, wind shear upwind of the domain box can generate a sloping aerosol feature within the time series; an apparent descending slope for aerosols below the jet max, apparent ascent for above the jet max. Khaykin et al. (2018) actually allude to this as a factor in the transport of the 2017 smoke plume. Considering that the smoke plume was transported from afar to the Asian sector, the role of wind shear in the transport and deformation should be acknowledged and investigated.

As a general matter, it has been shown in published results, of this case and other py-roCb stratospheric smoke plumes, that large meridional excursions of the plume from extratropics to subtropics and tropics is routine and not beholden to the AMA. Khaykin et al. (2018) show that for the 2017 event; their Figure 3 shows Canadian smoke south of 30N over the western Atlantic Ocean. Jost et al. (GRL, 2004) showed Canadian stratospheric smoke at subtropical latitudes. (In a paper under review, Fromm et al. extend the Jost et al. case study and findings to latitudes as low as 14N.) Fromm et al. (JGR, 2008) showed stratospheric pyroCb smoke at a tropical location (Hawaii). The path there did not involve nor require the AMA circulation. Pumphrey et al. (ACP, 2011) showed Australian stratospheric pyroCb CO in the tropical southern hemisphere. Sid-daway and Petelina (JGR, 2011) showed the tropical aerosol aspect of the CO plume that Pumphrey et al. presented. Hence the challenge for the present work is to con-vincingly show that the AMA was of consequence to the exclusion of (or together with) other demonstrable tropical plume excursions (E.g. Khaykin et al.'s Atlantic smoke).

Kloss et al. claim that there is no profile showing fire plume presence inside the AMA black box (Conclusions, P12, L14) but also infer (P6) that there is a SAGE smoke profile on 30 August inside that box. Their claim is at odds with Khaykin et al. (2018) who show (their Figure 3) CALIPSO plume detections well inside the black AMA box

on two dates in late August. Back trajectories that I calculated show that these plume segments connect with the synoptic-scale plume from a few days earlier over Europe, as shown in this paper (Figure 4) and Khaykin et al. (2018). This is seemingly at odds with the contention that the smoke plume bypassed the AMA center. Moreover, it is consistent with the general antecedent conditions of a large and expanding smoke plume advected from Canada to Europe to east Asia, including the region of the black box. Hence the big picture, as shown in this paper and Khaykin et al. (2018), is more in line with advective transport equally under the influence of all the flow regimes present throughout the northern hemisphere at that time.

---

## Author Comment (AC1) · 18 Jun 2019

*Referee #1*

*The authors would like to thank Reviewer 1 for the comments and ideas. We addressed each comment (black) below in blue in detail. Respective text changes in the manuscript are also indicated.*

1) The paper provides very interesting smoke observations (SAGE III, OMPS). As a suggestion, one could try to compare the observations with respective lidar observations (Hu et al., ACP, 2019, Haarig et al., ACP, 2018, CALIOP from September 2017 to March.

*We like the idea of comparing our satellite-based analysis with ground-based LiDAR measurements, which would make the study even more robust. However, both Haarig et al. and Hu et al. work with LiDAR measurements from Europe (France and Germany). The measurements in this study, which cover that region are shown in Fig. 1B. Unfortunately, with only 30 measurements per day, the SAGEIII data set does not provide a profile close enough to Europe end of August for any comparison.*

2) Regarding the simulations: In Haarig et al. (2018), they show a size distribution (the soot showed a pronounced aged accumulation mode, but no coarse mode) and they found single scattering values of 0.74 (355nm), 0.8 (532nm), and 0.83 (1064nm). In your simulation you use SSA of 0.9-0.93. This is quite high for soot! Any comment? Maybe, another simulation with SSA of 0.8-0.85?

*The interval SSA= [0.90-0.93] has been chosen, consistently with past observations/modelling of the evolution of fire plume optical properties (cited in our manuscript, see Sect. 4), as to mimic an aged fire plume. In this case, the plume is expected to be composed of "sulphate-covered soot" rather than pure soot. This is generally associated to SSA >0.9 rather than 0.80-0.85. Please note that our radiative calculations have been based on a plume 2-3 week older than the plume sampled in the work described by Haaring at el. (2018). Please also note that our hypothesis on SSA is quite consistent with Ditas et al. (2019), mentioned by Referee #1 (see comment 3). See in particular Fig. S12b of this latter paper. Basing on these considerations, we don't feel that a SSA of 0.80-0.85 would be representative of the plume at the conditions discussed in our manuscript; we also feel that adding the estimations based on a further group of simulations in the discussion of the radiative forcing of this plume, with pure-soot optical properties, would just be confusing for the Reader. If the Referee #1 still thinks that it can be useful, we might carry out new simulations and add this to Sect. 4. In any case, we added the reference to these observations of the plume in the revised manuscript: "This points at the presence of less absorbing features with respect to fresh biomass burning soot because of the progressive coating of condensed sulfates and/or organics (Ditas et al., 2019 and references therein). In addition, SSA for boreal forests fires have, on average, a higher SSA than tropical forests fires (Wong and Li, 2002).*

*The optical properties of this fire plume have been observed with a ground-based LiDAR, on 22 August in Europe, by Haarig et al. (2018). They report a SSA of 0.80 in the visible spectral range, which is typical of pure-soot particles. Nevertheless, our radiative simulations are representative of a plume at least 2 weeks older than the one sampled by Haarig et al. (2018) and with quite likely less absorbing (in terms of*

*absorption to scattering ratio) sulphate/organics-covered soot particles. Ditas et al (2019) have shown that SSA, for a biomass burning aerosol plume, is strongly dependent on the coating thickness of core black carbon particles. For aged fire plumes, a particle-to-core ratio of 4 or bigger was observed with in-situ aerosol observations on aircraft platforms (Fig. S12a of Ditas et al. (2019)). In these cases, the particles SSA has values of 0.90 or bigger (Fig. S12b of Ditas et al. (2019)). Therefore, we select 0.90 to 0.93 as the interval of SSA for the particular aged fire plume investigated in our paper".*

3) Please also check the paper of Ditas et al. (PNAS, 2018/19) concerning their simulation of the impact of soot on the radiation field.
*Ditas et al. is a very interesting study and relevant to this work. We added some discussion on our choice of SSA, which is supported by the findings of Ditas et al*

*Page 2 line 9: "Above southern France the plume was observed at altitudes up to about 20 km (Khaykin et al., 2018). Multiple studies have analyzed the fire plume above western/central Europe with LiDAR observations (Khaykin et al., 2018; Ansmann et al., 2018; Peterson et al., 2018). The general impact on the radiative balance and climate of aerosol plumes from wildfires in the lowermost stratosphere has recently been discussed in Ditas et al. 2018; they found that the global average direct radiative forcing at the top of the atmosphere (TOA) of biomass burning aerosols from wildfires may reach -0.20 W/m2 (including biomass burning plumes and biomass burning-affected background atmosphere, and including absorbing and scattering aerosol components)"*

*Page 10 line 31: "From the TOA RF calculations for the fire plume and ATAL, it can be concluded that the regional climate impact of the fire plume is up to 4 times (late ATAL, our estimation) and 2 times (peak ATAL, estimation by Vernier et al. (2015)) larger than the one of the ATAL. Our RF estimation for the fire plume is consistent with the estimated RF for biomass burning from wildfires of Ditas et al. (2019). The fire plume TOA RF estimated here in the tropical UTLS has the same order of magnitude as a moderate volcanic eruption. For example, Haywood et al. (2003) have estimated the mean RF…"*

---

## Author Comment (AC2) · 18 Jun 2019

*Referee #2*

*The authors would like to thank Reviewer 2 for the questions and suggestions made that help us make the paper overall more understandable and accessible for the Reader. Below, each comment (black) is addressed (blue) in detail, indicating the changes we have made on the manuscript.*

1) Page 6 starting from Line 15: ": : :in the whole NH >40N mid to end of August: : :" I am not sure I understand Fig. 1B, is there any data shown from mid to end of August 2017?

*We agree that this sentence is confusing, because the 'mid-to end- August bin' is actually missing for the SAGEIII data set with no measurements. We have changed the sentence to*

"A strongly enhanced aerosol extinction signature in the SAGE III data set is first visible in the beginning of September, in the whole NH >40N (Fig. 1B). Previous studies (Khaykin et al., 2018; Ansmann et al., 2018; Haarig et al., 2018) have shown increased aerosol extinction values associated with the fire plume from mid August."

2) "The aerosol signature descends with 2 mm in altitude per second" Could you please draw a line on Fig.2B to show the decline slope? It is hard to tell by eyeball. "Hence, the descent of the aerosol is due to sedimentation" Well, I believe sedimentation plays an important role. However, what about the longitudinal dilution and cross-latitude transport from higher latitude to lower latitude (not necessarily happen in ASM region)? How do those affect the 5 km/month rate derived in the manuscript?

*We added a decline slope (Fig. 1B) as requested.*
*We have changed the paragraph accordingly:*
"To exclude most cloud features and also background aerosol in Fig. 1B, we focus on the aerosol extinction region from ~0.6-0.9 km-1 for our analysis. A strongly enhanced aerosol extinction signature appears in the SAGE III data set, in the whole NH >40_N mid to end of August (Fig. 1B), after the beginning of the major fire event in Canada, confirming the results of multiple previous studies (Khaykin et al., 2018; Ansmann et al., 2018; Haarig et al., 2018). Between November 2017 and ~March 2018, the aerosol signature descends with 0.64 mm in altitude per second based on aerosol extinction values > 0.8 km-1 (5 km in three months, October to Janurary). This is in the order of the rate expected for the downwelling of the BDC (see Abalos et al.,2015). The effect of sedimentation is expected to play an important role. However, the contribution of sedimentation as well as dilution/mixing is not quantified here, microphysical and dynamical sensitivity studies would be necessary. The troposphere and lower stratosphere are filled with enhanced aerosols until mid-April 2018."

*To separate and quantify pure microphysical effects and dynamical ones, we would 1st need to compare simulations in one hydrostatic case (no transport) and 2nd include all processes (microphysics and dynamics). This would go beyond the scope of the presented study.*

3) Fig.1E: From OMPS, ATAL is mostly in troposphere; while from SAGEIII (Fig.2A,4A), ATAL's peak extinction is above 15km. Please explain why OMPS' ATAL is lower.

*The ATAL signal in the OMPS data set is visible up to ~18 km altitude (page 7 line 6, Figure 1E). In the SAGEIII data set (in Figure 2A and 4A) the ATAL signal has a maximum at 16 km. Hence, the agreement between both data sets for the ATAL height is very good.*

*Change in the manuscript: end of page 7/ beginning of page 8: '…in the wider Asian monsoon area (green box from Fig. 1A). Considering the data coverage of OMPS and SAGEIII, the ATAL height of the two data sets are in a reasonable agreement (up to 18 km for OMPS and peaking at 16 km for SAGEIII).*

*Looking at the green box… '*

*See also answer to comment number 6.*

4) Any stratospheric adjusting is taken into consideration in RF calculation especially for the 2017 fire plume with absorbing substance?

*Consistently with previous RF estimations for this event (e.g. Hu et al., 2019) no stratospheric adjustment is applied to our RF estimations. Nevertheless, this should have, in principle, a much smaller impact than for these previous studies, as we suppose a less absorbing fire plume (SSA=0.90-0.93) due to the longer atmospheric life when arriving to the AMA region than in Europe (see also Comment 2, Referee #1).*

5) In terms of equatorward transport of the plume, any other mechanisms/pathway can happen? For example, can the plume be lifted higher in the stratosphere in mid-hight latitudes, and then been transported to the tropics? Is there any way you can quantify/ compare the relative fraction of plume transported to tropics via the two ways respectively?

*Quantifying the exact transport of the fire plume of the Canadian wild fires in 2017 into the stratosphere is an interesting subject. Part of the fire plume was directly injected into the stratosphere above the fires, then transported within the jet to the Asian monsoon region. However, it is possible that part of the fire plume was first transported within the troposphere and later uplifted into the UTLS. Some of the coauthors are working on a different study to explore this further.*

6) Fig.4 I am a little lost here: a. In 4A, which is ATAL and which is fire?
b. Is CO in 4B and 4C associated with fire at all? From Fig.3, authors suggest that ASM barrier prevents fire smoke mixed in; If CO can be mixed in, why not aerosols?

*The peak in Figure 4A at 15-16 km altitude can be associated with the ATAL and the peak at 17-18km with the fire plume.*

*Enhanced CO values in Figure 4B are associated with the 'general' enhancement of tropospheric tracers inside the AMA and do not originate from the fire plume.*

*We realize that the corresponding sentences lack some clarifying details (starting line 33 page 8). Therefore, we have changed it to: "A measurement profile with readily identifiable and vertically separated AMA and fire plume signatures is shown in Fig. 4A. At around 370 K, the profile inside the AMA shows no clear evidence of enhanced aerosol from the fire plume (Fig. 4 A at ~16km altitude). This is consistent with the*

*existence of the generally strongest confinement (transport barrier) at around 380 K (Ploeger et al., 2015). The generally weaker confinement at around 400 K compared to 380 K is reflected by the CO gradient and Montgomery stream function shown in Fig. 4B. The enhanced CO mixing ratios displayed in Figure 4B, indicate the entrainment of tropospheric tracers insider the AMA (e.g. Santee et al 2016, Park et al. 2006).*

*The profile in Fig 4 is selected here, because of its location within the eastern flank of the AMA circulation (Fig. 4B), within the canonical northsouth transport pathway from the extra-tropics to the tropics. Back-trajectories show that air masses from the altitude levels of the fire plume and ATAL peaks pass over partly different regions 9 days prior to the SAGE III measurement profile (Fig. 4C). For this study ultra-violet (UV) aerosol index measurements by OMPS and the position of detected enhanced aerosol extinction values by CALIPSO are displayed in Fig. 4C. While CO mixing ratios are a fire indicator for 'fresh' plumes, enhanced aerosol can be traced over longer time scales. Because of the spatial distance between the fire plume origin and the Asian monsoon region, aerosol extinction values rather than CO measurements are taken as an indication for the fire plume…."*

7) Minor: Fig.3, why there are 2 identical color bars?
*We replaced the two color bars by one color bar.*

---

## Author Comment (AC3) · 18 Jun 2019

*M. Fromm*

*We would like to thank Michael Fromm for the detailed questions and comments made. As a general remark: we do not contradict other possible transport mechanisms of various fire plumes into the tropics, but rather analyze this specific case of a far northern fire and the fast transport pathway within the jet to the Asian monsoon region and into the tropics. We will emphasize this issue better to avoid misunderstandings.*

1) Abstract. In the first sentence (": : :reached the tropics, and subsequently the tropical stratosphere: : :") Kloss et al. seem to suggest that the Canadian smoke plume, upon entry into the area of the AMA, had a discernible tropospheric component. Only subsequently was it lofted into the stratosphere by the BDC according to this claim. This is a fairly provocative claim. However I could not find any evidence given or figures showing upper tropospheric smoke adjacent to and wrapping around the AMA. They attribute all the aerosols displayed below the tropopause to the ATAL. The evidence in Figure 1 and Khaykin et al. (2018) shows that by late August the smoke near the AMA was already at stratospheric heights and potential temperatures. If my understanding of the claim set forth in the abstract is correct, to defend it would require two things. 1. an unambiguous discernment of upper tropospheric smoke upstream of the tropical observations, and 2. evidence ruling out quasi-isentropic transport of the observed stratospheric smoke to the tropics. If on the contrary it is acknowledged that the smoke moving into Asia in late August was already spanning the lower stratosphere (as Khaykin et al. (2018) show) then it is hard to defend the abstract's claim convincingly.
*We agree that this sentence is misleading. We really only analyze the fire plume signature that already reached the Asian monsoon area within the stratosphere. We emphasize the 'tropical stratosphere' because there it has the potential to be uplifted within the BDC and reach the 'global' stratosphere. We do not want to indicate here that we analyze any tropopause crossing into the stratosphere.*
*The respective sentence in the abstract was chaned to: "We show that a fire plume injected into the lower stratosphere at high northern latitudes during the Canadian wildfire event in August 2017 reached the tropics, and was subsequently further lifted in the tropical stratosphere within the ascending branch of the Brewer-Dobson Circulation (BDC). "*

2) On a technical but important note, the Abstract mentions "July" as part of the Canadian smoke event. There is no evidence here or in other papers that July was in play. This wording should be removed.
*Ok, the corresponding sentence was changed (see answer to comment 1).*
3) Introduction, L2-3. The manuscript stipulates that pyroCb activity is the source pathway for this plume. Hence it is critical to accurately establish the pyroconvective source. That is best done by citing Peterson et al. (2018) in this sentence. Peterson et al. give detailed and accurate constraints on both the pyroCb injection in the Pacific Northwest and the 3D footprint of the pyroCb plume on 14 August. Khaykin et al. (2018) points the reader to fires that did not exhibit pyroCb activity. (Sergey and I have had a personal communication on that matter.) Hence that paper is not fitting as a citation here.

*Thank you! We changed the citation to Peterson et al.*

4) On that topic, the choice of initializing CLaMS over three days centered on a box that is neither focused on the Pacific Northwest pyroCbs nor the pyroCb plume on a subsequent day seems destined to introduce many spurious or useless trajectories. The growing realization that there was significant diabatic lofting of the smoke further diminishes the applicability of the CLaMS construct and setup. Consequently little confidence can be gained from a set of these trajectories at a single potential temperature surface (especially since the plume was lower than 380 K in the first days (See Fig. 4 of Khaykin et al. (2018)).

*The CLaMS simulation is initialized at the time and altitude level of observed enhanced CO values (IASI measurements) due to the fire (see caption of Fig. 3). From the CLaMS simulation we do not derive any quantitative results of 'how', 'when' and 'how much', but rather use it as a qualitative (2D) visualization of the estimated transport pathway. We agree with M. Fromm that our initialization could also tag some air masses outside the IASI CO plume, although the box was chosen around the observed plume. To address the robustness of the deduced transport pathway we also initialized air masses (inside the same horizontal box) on each single day between 12th and 14th August, and on the different potential temperature levels between 345 and 465K.The transport pathway via the Asian monsoon circulation into the tropics emerged very robustly from all these sensitivity experiments. Therefore, for the paper we decided to show the transport of the total mass tracer tagging all air masses in the Canadian box for 12.-14. August and the entire layer 345-465K.*

*For clarification, the respective paragraph has been modified:*

*"To investigate the dynamics of the fire plume transport to the AMA region, an air mass origin tracer has been initialized between August 12th and 14th 2017 in the box over western Canada (green box in Fig. 3), using the CLaMS model.The point in time and space of the initialization box was chosen according to the position and time of high observed IASI CO vaues due to the fire. The simulation with box initialization as presented here, is a good indicator for possible large-scale transport pathways, but however should not be taken for quantitative estimations as some air masses within the box could not belong to the fire plume. The model fire tracer was injected in the respective box throughout the layer 345-465K, as observed by IASI . This approach was found to be very robust, by initializing air masses on different potential temperature levels (345-465K) and on each day between August 12th and 14th. Therefore, uncertainties arising from the observed time and injection altitude do not interfer with our line of arguments. After initialization, the tracer has been advected passively during the following weeks. This approach is similar to the one presented in (Vogel et al. 2015): the plume is first transported eastwards, at latitudes >40_N and passes over Europe in early/mid-August (Fig. 3A). After reaching the Asian monsoon area at the end of August, a fraction of the fire tracer is partly transported along the eastern flank of the AMA circulation from the extratropics into the tropics (Fig. 3B). In the simulations, part of the plume even reaches the southern hemisphere (Fig. 3C). It is shown that the plume reaches the tropics (<10_N) first through the AMA circulation (Fig. 3C). This is consistent with the SAGE III observations shown before. With the slow breakdown of the AMA, plume air masses mix into the area that has before been confined by the AMA transport barrier from the northern side (Fig. 3D). By mid-September most of the NH is*

*filled with the artificial fire tracer at 380 K potential temperature (Fig. 3D). This pathway of the fire plume transport to the tropics within the eastern flank of the AMA circulation is further confirmed by OMPS aerosol extinction observations (see Figure S2 of the supporting material)."*

5) Introduction, L29. Of the 3 papers cited on this line, only one postulates the Nabro troposphere-ASM -convection pathway: Bourassa et al. (2012). Fairlie et al. Dispute that claim. Sellito et al. seem to be noncommittal on the pathway. Considering that Kloss et al. are apparently attempting to draw parallels with the Nabro publications and the 2017 AMA/smoke interaction (P2, L30), it is important to accurately portray the literature on the Nabro event.

*The respective sentence was changed to: "For the Nabro volcano eruption, for example, the emitted aerosol and precursors have been partly injected directly into the lower stratosphere (Vernier et al. 2013, Fromm et al., 2013) at altitudes of about 15-18 km (Clarisse et al., 2014, Fromm et al., 2014). It has been suggested that a fraction might have been transported into the stratosphere via the upwelling in the Asian monsoon (Bourassa et al., 2012, 2013). Satellite observations of volcanic effluents as SO2 (Clarisse et al., 2014) and sulphate aerosols (Sellitto et al., 2014) have shown the interaction of the plume horizontal dispersion and the AMA dynamics."*

6) P3, L22. Why was it decided to use "cloud unfiltered" SAGE 3 data? Thomason and Vernier (ACP, 2013) were compelled to go to great lengths to adopt a rigorous cloud clearing in SAGE II data for the study of tropospheric aerosols (indeed the ATAL). For inadequately constrained data sets such as SAGE and OMPS it is essential to either attempt aerosol-cloud discrimination or acknowledge that the tropospheric information content is uncertain. This is especially true for a regime like the particularly cloudy ASM.

*We have originally done both (filtered and unfiltered) and actively decided among the coauthors to use the unfiltered version. We have decided for the unfiltered version, because we focus on the fire plume signature near the tropopause and the conclusions drawn were the same (+ the filtering process by Vernier and Thomason did not remove all cloud-like features in the new data product of SAGEIII). Note that newly cloud-filtered SAGEIII data are currently developped (Jean-Paul Vernier, personnal communication). The OMPS data are 'cloud-filtered' (only data above the 'top of cloud'-altitude are taken).*

7) P6, L17. Like one of the reviewers, I do not see evidence of descent. In fact it can be argued from this figure that aerosol is ascending. Indeed Khaykin et al. (2018) show that the extratropical smoke plume height increased dramatically, presumably due to diabatic forcing. What is the indicator of descent?

*"To exclude most cloud features and also background aerosol, we focus on the aerosol extinction region from ~0.6-0.9 km-1 for our analysis. A strongly enhanced aerosol extinction signature appears in the SAGE III data set, in the whole NH >40_N mid to end of August (Fig. 1B), after the beginning of the major fire event in Canada, confirming the results of multiple previous studies (Khaykin et al., 2018; Ansmann et al., 2018; Haarig et al., 2018). Between November 2017 and ~March 2018, the aerosol signature descends with 0.64 mm in altitude per second based on aerosol extinction values > 0.8 km-1 (5 km in three month, October to Janurary). This is in the order of the rate expected for the downwelling of the BDC (see Abalos et al.,2015). The effect of sedimentation is expected to play an important role. However, the contribution of*

*sedimentation as well as dilution/mixing is not quantified here, microphysical and dynamical sensitivity studies would be necessary. The troposphere and lower stratosphere are filled with enhanced aerosols until mid-April 2018."*
*We have added a line to guide the eye in Fig 1B and D, as also suggested by one of the reviewers.*

8) P6, L20. I don't see any difference in the extinction pattern after mid-April as compared to just prior to mid April. In fact tropospheric extinction appears to be saturated red throughout the timeline. I refer back to my comment above regarding cloud contamination and suggest that it is not possible to argue that the preponderance of the unfiltered tropospheric extinction signal on display is from aerosol.
*We agree that this paragraph as written is not clear. We have added a sentence in the beginning (see answer to comment 7) to clarify what aerosol extinction range we focus at.*

9) P6, discussion of Fig. 1C. The value and information content of this figure panel is not obvious. As the authors state, detailed interpretation of smoke layers is hindered by the lack of filtering. In addition, half of the period rendered is the winter season, when there is no anticyclone and confinement. Presumably smoke aerosols would be in evidence in any other longitudinal sector in the winter. Hence some additional explanation of the meaning of Figure 1C is called for meaning of Figure 1C is called for.
*We believe this plot is very important and have added another statement about the arrival of the plume already in the lower stratosphere.*
*"Fig. 1C shows the SAGE III aerosol extinction values in the inner AMA region (black box in Fig. 1A). The unfiltered cloud structures in the SAGE III data set masks the first appearance of the plume in the back box in Fig. 1C. However, the first SAGE III profile that we can track back to the fire plume signature originating from the Canadian wildfire appears on August 30th 2017 at 17 km altitude. The relatively high altitudes of this signature (17-20 km) indicate that the fire plume arrived in the TTL region in the Asian monsoon area, where the upward motion inside the AMA might have forced the fire plume to rise, as it was the case for the Sarychev aerosol plume in 2009 (Vernier &,Thomason 2011). A clear signal is still apparent in April 2018, 8 months after its first appearance and long after the break down of the AMA confinement. However, it has to be noted that there are no previous years of SAGE III measurements available so that no comparison with background conditions in April can be made."*

10) P6, L29. Like the discussion of descent earlier, it is not evident what feature suggests ascent in Figure 1D. Moreover, there are additional plausible explanations for a sloping aerosol feature in a time series set in a localized domain. For instance, wind shear upwind of the domain box can generate a sloping aerosol feature within the time series; an apparent descending slope for aerosols below the jet max, apparent ascent for above the jet max. Khaykin et al. (2018) actually allude to this as a factor in the transport of the 2017 smoke plume. Considering that the smoke plume was transported from afar to the Asian sector, the role of wind shear in the transport and deformation should be acknowledged and investigated.

*The respective paragraph in the manuscript has been modified: "To see whether the fire plume has entered the AMA circulation and has been transported to the tropics (as it has been shown for the Sarychev eruption by Wu et al. (2017)), another box south of the core Asian monsoon box has been chosen (Fig. 1A, magenta box). We attribute the ascending signal starting at around 16km in mid September and reaching altitudes of around 21 km about 6 months later to the Canadian wildfire, as its origin coincides in time and altitude with the fire signal in the AMA region (black box, Fig. 1C). In the tropics, the fire plume signature rises about 0.2-0.3 mm per second (about 5 km from September to April) in the magenta box according to aerosol extinction values of around 0.6 km-1. This tropical upwelling velocity estimate is in good agreement with the tropical upwelling velocity in current reanalyses (e.g., Abalos et al., 2015, Fig. 6). Similar ascending features are visible around the globe 0-25C.*  *The reversed vertical transport of the aerosol particles in Fig. 1B compared to 1D (i.e. the observed descent in the northern latitudes and ascend in the tropics) reflects the contribution of the ascending and descending branch of the BDC. The average signal for the magenta box remains also until April 2018 at ~19 km altitude. The AMA generates a strong connection between the mid-latitudes and the tropics during the summer season."*

11) As a general matter, it has been shown in published results, of this case and other pyroCb stratospheric smoke plumes, that large meridional excursions of the plume from extratropics to subtropics and tropics is routine and not beholden to the AMA. Khaykin et al. (2018) show that for the 2017 event; their Figure 3 shows Canadian smoke south of 30N over the western Atlantic Ocean. Jost et al. (GRL, 2004) showed Canadian stratospheric smoke at subtropical latitudes. (In a paper under review, Fromm et al. extend the Jost et al. case study and findings to latitudes as low as 14N.) Fromm et al. (JGR, 2008) showed stratospheric pyroCb smoke at a tropical location (Hawaii). The path there did not involve nor require the AMA circulation. Pumphrey et al. (ACP, 2011) showed Australian stratospheric pyroCb CO in the tropical southern hemisphere. Siddaway and Petelina (JGR, 2011) showed the tropical aerosol aspect of the CO plume that Pumphrey et al. presented. Hence the challenge for the present work is to convincingly show that the AMA was of consequence to the exclusion of (or together with) other demonstrable tropical plume excursions (E.g. Khaykin et al.'s Atlantic smoke).

*Those papers are relevant to our work and will be mentioned.*
*We do not contradict other possible transport mechanisms into the tropics. We want to emphasize that we do not exclude the possibility that fire plume aerosols can also occur in the tropical stratosphere without any Asian monsoon anticyclone circulation interaction. Of course, the location of the occurring fire event is highly sensitive to the following transport mechanisms and also the time scale. In this case study we focus on the fire plume that was transported within the jet, reaching the Asian monsoon region and then transported around the anticyclone. We do, however, show for the first time that a fire plume originating from northern latitudes is transported within the circulation of the AMA (while not interacting with the isolated center of the AMA). Figure 3 by Khaykin et al. (2018) is limited to August 2017 and ~30°N while our Figure 1 and 3 (and discussions) focus also on September and the following months. From this point of*

*view, the Khaykin study does not contradict our work, but is rather taken as an input location from where part of the plume is transported around the anticyclone (e.g. see Figure B2 of the supplements).*

12) Kloss et al. claim that there is no profile showing fire plume presence inside the AMA black box (Conclusions, P12, L14) but also infer (P6) that there is a SAGE smoke profile on 30 August inside that box. Their claim is at odds with Khaykin et al. (2018) who show (their Figure 3) CALIPSO plume detections well inside the black AMA box on two dates in late August. Back trajectories that I calculated show that these plume segments connect with the synoptic-scale plume from a few days earlier over Europe, as shown in this paper (Figure 4) and Khaykin et al. (2018). This is seemingly at odds with the contention that the smoke plume bypassed the AMA center. Moreover, it is consistent with the general antecedent conditions of a large and expanding smoke plume advected from Canada to Europe to east Asia, including the region of the black box. Hence the big picture, as shown in this paper and Khaykin et al. (2018), is more in line with advective transport equally under the influence of all the flow regimes present throughout the northern hemisphere at that time.

*The sentence says 'inside the AMA' and not 'inside the AMA black box'.*
*Our sentence (P12 L14) "There is no profile showing that the fire plume passes the barrier, mixing with the air masses inside the AMA." is true and important (one of the main messages of the paper). It is neither at odds with Khaykin 2018, nor with the plume being above Europe a few days before. There are several profiles inside the transport barrier of the AMA with no fire signature (e.g. see Figure B3 of the supplements, with the ATAL signal).*
*The black box from Figure 1 is chosen for a statistical approach, showing that this is an area of mostly being inside the AMA. The SAGEIII profile of the 30th of August is analyzed in detail and it is shown that this profile (on that particular day and on that particular altitude level) is within the flow of the anticyclone and not within the transport barrier.*

---

## Referee Report (RR1)

Upon the original submission of Kloss et al (hereafter K19) I offered comments in the capacity of a generally interested scientist, not a reviewer. Having been named a reviewer of K19's revisions, my review will begin with an overall statement, followed by comments on K19's reply to the two reviewers and me, then a presentation of additional comments on the current manuscript.

K19 present a mostly Asian Monsoon centric analysis of a smoke plume generated in western Canada in mid-August 2017. Their objective is twofold, to show how the Asian Monsoon Anticyclone (AMA) mediates the transport of aerosol into the tropics and to compare radiative forcing of the smoke plume with particles in the Asian Tropopause Aerosol Layer (ATAL). To do so K19 use SAGE III ISS and OMPS/LP aerosol extinction profiles to characterize both the stratospheric smoke plume and the tropospheric aerosols. A third prime aspect of this paper was to assess vertical motions of the smoke, both in the extratropics in general and tropics at AMA longitudes.

I find the original and revised manuscript to be unconvincing owing to a flawed analysis construct and confusing presentation. The revisions did not adequately deal with the faults which I found –with an important exception mentioned next-- in the original submission. Moreover, K19 failed to adequately address some major points made in the review comments. Moreover they also indicated they would make some changes that were not in the revised manuscript. Hence my assessment is that this manuscript is not acceptable in its present form; it will require wholesale changes in order to reach their stated objectives and ACP standards.

On one point K19 succeeded in addressing one of my core concerns, which was that the original manuscript led the reader to understand that there was a component of troposphere-to-stratosphere transport in the AMA's mediation of the smoke plume. It is acknowledged that the revised manuscript appropriately and clearly constrains the smoke plume component of the analysis to the stratospheric realm.

K19's reply to Reviewer #1: no issues.

K19's reply to Reviewer #2: no issue except for concerns that overlap with mine. They will be discussed in the context of the replies to me.

K19 Reply to MDF

In their reply to reviews and short comment, K19 numbered each target comment in review order. I will use this numbering convention in my critique below.

One) This is where my concern about troposphere-to-stratosphere evolution was brought up and responded to. K19 are to be acknowledged for making the suggested changes.

Four) This regards the concern that K19's CLaMS initialization box, altitude range, and dates introduced many unrepresentative start points for the transport simulation. K19 acknowledged that spurious trajectories would be introduced but argued that in fact using these larger-than-defensible constraints made the results more robust. I fail to see how adding demonstrably unrepresentative initial conditions adds robustness to a model simulation. For example, the pyroCbs occurred on 12 August so the first representative IASI CO plume wouldn't have shown up until 13 August. So every trajectory launched on 12 August is unrepresentative. The CO plume covered only a small fraction of the box they chose, so every data point outside the CO plume is unrepresentative. The CO plume changed in position between

13 and 14 August, so different unrepresentative trajectories were launched on those two days. We know from Khaykin et al. that the initial plume was below 380 K, so every trajectory above the plume was unrepresentative. Finally, as pointed out in my comment, Khaykin et al. showed unequivocally that the smoke plume ascended diabatically from below to higher than 380K in the first month after the pyroCbs. So choosing a single theta to demonstrate robustness is physically inappropriate. The results shown in their Figure 3 are needlessly numerous and cover so much of the NH that many conclusions can be drawn from them that have no bearing on the AMA. Moreover, there are many tongues of trajectories into the tropics at longitudes far from AMA influence that are not attended to. Hence I find this analysis to offer little value at best, and probably misleading at worst.

Five) A minor matter regarding Nabro and Bourassa et al. K19 did improve the wording here but two items of concern remain. One is that the text makes a false statement, that Bourassa et al. argued that a "fraction" of the Nabro stratospheric plume came up from the troposphere. In fact Bourassa et al. claim that the entire injection was well below the tropopause, 9-14 km, hence 100% of the eventual Nabro stratospheric plume came from the troposphere. Secondly, since K19 have correctly clarified that the focus of this work is on a stratospheric plume, there is no apparent justification for this discussion of the Bourassa claim in this paper. To me it deflects from K19's premise and weakens the introduction.

Six) Regarding filtering of limb extinction profiles. This is a critical concern. To the extent that K19 use extinction profiles to characterize aerosol burden in the UT, their answer is problematic. On one hand K19 state that even the strategic attempts by Thomason and Vernier to remove cloud data from SAGE UT extinction did not remove all cloud-like features while on the other hand state that they opted for unfiltered data, which to me suggests that the resultant data set would be even more compromised by cloud. The upper troposphere in the summer Asian Monsoon region is one of the cloudiest places on Earth. Hence unfiltered extinction data in this region are bound to represent thin or subvisual cirrus much more than tenuous and uncertain aerosol layers. K19's reply that their conclusions were the same between tests with filtered and unfiltered profiles can be interpreted as meaning that upper tropospheric cloud statistics mimic an unknown aerosol signal. Consequently, to present their results as a signal of the ATAL is an oversimplification at best, and significantly cloud- biased at worst. K19's use of the OMPS cloud height to filter these data does nothing to improve the confidence that tropospheric extinctions above the cloud height are only attributable to aerosol. The same thin or subvisible cirrus will impart marginal increases to extinction even above the OMPS algorithm's cloud height. In summary, tropospheric extinctions by themselves are woefully under constrained for separating aerosol from cloud.

Seven) Regarding Figure 1's indicators of descending and ascending aerosols. This is also a critical item. K19 respond to my comment and one of the reviewers by adding a line to the North_of_40 and tropical Asia figure panels (1b and 1d, respectively). While visually this is an appropriate response, scientifically it has no weight. There is no physical argument presented by K19 or previous research (to my knowledge) for use of a single extinction contour to map out a phenomenon such as sedimentation or BDC-driven vertical velocities. Even in the specific context of Figure 1 there is no apparent basis for these elections. In the case of the North_of_40 time-height plot, there are different slopes with different extinction choices, even slops that are of a different sign. The fact that the smoke plume extends much higher than the altitudes of the slope marker indicates that material vertical movement within that plume envelope is impossible to isolate with confidence. There are many reasons for such sloping patterns that owe nothing to the BDC. For all these reasons, this slope analysis and interpretation thereof has little merit.

Seven) Same MDF comment, but regarding Khaykin et al.'s finding of diabatic ascent within the smoke plume. K19 did not address this apparent discrepancy between one of Khaykin's findings regarding the same plume within which K19 interpret descent. K19 did not address this critical point.

Eight) Regarding a specific sentence that makes a broad characterization of tropospheric and stratospheric aerosols filling the scene through April 2018. K19 did not address this point directly. The question I raised, which goes to my concerns w.r.t. the absence of SAGE extinction filtering, stands.

Nine) Regarding Figure 1c and K19's interpretation. K19's reply did not address the concern. The original and revised text refers to a SAGE III "profile" that can be connected to the Canadian source, but Figure 1c does not show any individual profile, only smoothed averages. They specifically call out an individual SAGE profile on 30 August but it is not in Figure 1c nor anywhere else to which they refer the reader. Apparently responding to this concern, K19 inserted a sentence that ostensibly refers to this specific profile's aerosol enhancement ("The relatively high altitudes of this signature (17-20 km) indicate…") but they still did not refer the reader to a plot of the 30 August SAGE profile. The inserted text invokes the TTL and "upward motion inside the AMA" forcing the plume to rise. The revised text is altogether vague, confusing, and hence immaterial as a revision.

Ten) Regarding the claimed "ascent" of smoke in the tropics exhibited in Figure 1D. K19 made a revision to the figure (a sloping line) and text analogous to their change to the Noth_of_40 plot in Figure 1C. For the same reasons expressed in 7) I don't give merit to this slope. In my comment I expressed the assertions of Khaykin et al. that provide an alternate explanation for any positively sloping feature, but that was not addressed by K19.

Eleven) Regarding previous literature on pyroCb smoke plumes entering the tropics. K19 acknowledge the relevance of these papers and state that they will be mentioned. The revised paper has no such discussion nor citations.

Eleven) K19 argue that they "do not contradict other possible transport mechanisms…" in an apparent misunderstanding of my concern. My issue isn't that K19 contradicts other transport explanations but rather ignores the large weight of evidence that the AMA is simply one of several interconnected dynamic drivers of meridional transport in the lower stratosphere. It was to that purpose that I mentioned the background literature on the subject. My concern stands, and I contend that K19 need to adopt a less AMA-centric construct in order to prove its role in mediating the transport of this smoke plume.

Twelve) Regarding SAGE stratospheric smoke observations in the AMA box. This is a critical item. My comment was triggered by what I found to be confusing discussion centering on Figure 1a and 1c. K19 establish a box (the black box) to mark the core of the AMA, the area with the greatest probability of air-mass confinement during the summer. In K19's discussion (P7, L5-13 in the original manuscript), they discuss the SAGE statistical view of extinction in the "inner AMA region." They then immediately refer to an individual SAGE profile on 30 August (presumably inside the AMA black box) that they can trace to the fire source, but that profile is not in Figure 1c, and K19 do not refer the reader to any other figure at this point in the discussion. Hence I found it impossible to know whether K19 was referring to the smoothed features in Fig. 1c or something else that wasn't displayed. Hence the points made in this paragraph were impossible for me to process. Even so, it was clear that K19 were discussing the inner AMA region and noting that there was a smoke layer on 30 August inside that region. That is why the

statement in the Conclusions seemed at odds with the discussion of Fig. 1c. The profile on 30 August to which K19 presumably refer is shown in an appendix. This profile, at ~22N, is outside the black AMA box that presumably is the focus of discussion regarding Fig. 1c. (Additionally, the coordinates in the figure caption do not match the mapped and plotted profile.) This profile seems to be of central importance (since K19 called it out and analyzed it in detail) hence the figure needs to be called out when it is referred to, and ideally it should not be shunted an appendix. Given that this profile is outside the black AMA box, it may not even be the profile to which K19 are alluding in the discussion of Fig. 1c. Perhaps K19 intend to refer to the profile in Fig. 4, which is on 31 Aug, not 30 Aug. All things considered, this analysis, which is of central importance to the paper, is immensely confusing to me.

Revised Manuscript

Substantial issues are itemized here. Technical/minor items are provided as comment bubbles in the pdf of the manuscript.

Section 3, P6L29-P7L04, discussion of Figure 1B showing the "whole NH>40°N." This misleads the reader by suggesting the >40N zone comprises SAGE sampling of the whole NH. There are no SAGE profiles north of 50N for almost the entire winter half of the year. So effectively the >40N zone is a very narrow midlatitude belt. The fraction of the whole NH that is effectively sampled is quite small, considering the additional fact that there is a huge extratropical area of the NH south of 40N. See the SAGE III latitude-time map below. Bold symbols are north of 40N. The horizontal bars bracket the southern limit (40N) and the mid-winter northern SAGE III profile extent.

[Figure]

Also important to note is SAGE's temporal latitudinal sweep. This effectively narrows the meridional representativeness of these data for the case of a rapidly evolving, inhomogeneous phenomenon such as a weeks old smoke plume. It is readily seen here that the aerosol enhancement features in Figure 1B likely have imprints of many factors dominating the BDC or any other singular explanation. Even if there is a physically meaningful slope in Figure 1B, it is not necessarily synonymous with downwelling. This box is effectively less than 10 deg. in latitude, considering that SAGE III did not measure north of 50N for most of this period. So it simply represents a narrow belt within which to observe smoke that came from north of the belt and almost certainly moved meridionally in and out of the belt in both directions over these several months. n short, the SAGE III data shown in Figure 1B do not represent the "whole NH>40°N," which would be the necessary condition for permitting K19's interpretation of Figure 1B.

Section 3, P7L14-P8L09, discussion of Figure 1D (tropical monsoon sector). This comment relates to my original comments (see 7) above. A slope such as shown in Fig. 1D is not synonymous with upwelling. Upwelling refers to a process that cannot be discerned in a box of such small dimensions. Considering mean UTLS wind patterns and speeds in this box, parcels of air (and hence individual particles) do not reside in this box for more than a few days. The following two plots of back and

forward trajectories initialized in the magenta box show that the aerosol content is a transient; the source and fate of any aerosol measured in the box extends far beyond the confines of the box.

[Figure]

[Figure]

NOAA HYSPLIT MODEL
Forward trajectories starting at 0000 UTC 01 Nov 17
GDAS Meteorological Data

Job ID: 120685    Job Start: Wed Jun 26 14:23:20 UTC 2019
Source 1    lat.: 5.000000    lon.: 40.000000    height: 19000 m AMSL

Trajectory Direction: Forward    Duration: 240 hrs
Vertical Motion Calculation Method:    Model Vertical Velocity
Meteorology: 0000Z 1 Nov 2017 - GDAS1

Ten-day back and forward trajectories launched from a matrix within K19's magenta box, on 1 November 2017 at 19 km. Most parcels in the box were not there 10 days before or 10 days after. Hence to discern a process such as BDC-forced upwelling a more global analysis is required.

P8, L04. "Similar ascending features are visible around the globe 0-25_N." This sentence was added to the revised manuscript. This statement is given without any support. If K19 observed such an important feature, it merits inclusion as a figure. However, even if that global tropical slope is provable, it does not diminish the argument that forces other than BDC upwelling require consideration. The plume originated in the extratropics; its pre-tropics disposition and its flux into the tropics are essential considerations. These considerations include recognizing the reported diabatic lofting of the extratropical plume by Khaykin et al. (2018).

P8,L10-18. Discussion of Figure 1E. Here K19 draw the reader's attention to the smoke plume in the stratosphere and what they conclude to be the tropospheric ATAL. It's not clear what connection is being made here. K19 have stipulated that the smoke plume they're examining is stratospheric, hence its relation to the ATAL feature, which occurs annually, is not explained clearly. Please elaborate or consider removing this paragraph.

P8L10-18. Another question about this paragraph. Earlier in the paper K19 establish, using Figure A1, that the AMA is still around at the end of September. Presumably there is some area of confinement inside the AMA black box until then. Yet OMPS/LP smoke aerosols are found in the black box: "The fire plume appears…at the beginning of September 2017." In light of K19's concluding statement about no smoke aerosols crossing the mixing barrier, further clarification of arguments made in this paragraph are sought.

Appendix B. P16L08-L31.Discussion of Figure B2. To me the trajectory analysis is ambiguous and the discussion here is confusing. For all intents and purposes, the blue and red trajectories show identical histories in 9 days. The only visible difference is that the red ones are a little longer (the wind speeds are generally a little greater). Even the curling of the blue trajectories in the southeast part of the history is shared by the red ones (they just look less obvious because they are plotted underneath the blue lines). K19 refer to 21 August and western China regarding the red trajectories, but the red trajectory termination on 21 August is far from China, over central Asia. It is unclear how Khaykin's observation in China on 21 August applies here. It is also unclear what is meant by "the first fire plume signal" in regard to Khaykin; clearly Khaykin reported on fire plume observations prior to 21 August. As a consequence of this confusing discussion, the only thing I can take from the figure is that the air reaching the SAGE profile on 30 August at both altitudes was indistinguishable history-wise.

Appendix B, P16L32. "Another clear example for the fire plume signal measured in the Asian region in given in Fig. B2." This sentence starts a paragraph that seems to be discussing Figure B3, not B2. But B3 shows only what K19 are calling the ATAL. I'm confused. Please reconcile the figure callouts and the messaging here.

[revised manuscript text omitted]

---

## Author Response (AR2)

**Reply to MF**

We thank MF for his comments on the manuscript. We see this criticism as a challenge to further improve the paper and tried to thoroughly address all points. Although some of the review comments are severe, we think that they are mostly related to misunderstandings and can be addressed by improving the clarity of the presentation. Therefore, we substantially modified the text and included new Figures (Fig. 4, Fig. C1 and Fig. D1). The structure of the responses was chosen according to the MF review. Changes made to the manuscript are indicated in red.

First of all, after the ACP open discussion phase of our paper, another study was published on a very similar topic, of which MF is a coauthor (Yu et al., 2019).

With the newest addition of relevant literature by Yu et al. 2019, we have modified the introduction accordingly:

*'During the 2017 summer season, historically severe wildfires appeared in western Canada and in the north-western United States. Strong thunderstorms (pyro-cumulonimbus activity), which developed above the fires, injected smoke particles above western Canada into the lower extratropical stratosphere in mid-August (Khaykin et al., 2018). The fire plume was transported through the jet stream eastward and rose 2-3 km per day within the first days after its injection into the stratosphere (Khaykin et al., 2018)* and to an altitude of 23 km within two months (Yu et al., 2019). *Three days after the first appearance in the stratosphere above Canada, the plume first appeared over Europe on August 19th. Above southern France the plume was observed at altitudes up to about 20 km (Khaykin et al., 2018). Multiple studies have analyzed the fire plume above western/central Europe with LiDAR observations (Khaykin et al., 2018; Ansmann et al., 2018; Peterson et al., 2018).* Yu et al. 2019 recently showed with model calculations that most of the fire plume in the stratosphere was quickly transported to the poles (depending on the computed BC content). *The general impact on the radiative balance and climate of aerosol plumes from wildfires in the lowermost stratosphere has*  *been discussed in Ditas et al. (2018); they found that the global average direct radiative forcing at the top of the atmosphere (TOA) of biomass burning aerosols from wildfires may reach -0.20 W/m² (including biomass burning plumes and biomass burning-affected background atmosphere, and including absorbing and scattering aerosol components).'*

**FOUR)** The key point of the paper is really the observations of the plume in the AMA region.

-We already provided explanations in the previous reply to MF comments (as well as in the manuscript, see below). The initialized (green) box over western Canada is based on IASI **observations**. The simulation is in fact a **3D simulation**, displayed as a 2D plot. We argue that it is possible that CLaMS tags along some air masses which are not filled with fire, but that is due to the box like shape of the initialization region. As we don't have a continuous description of this plume, a box shape which is a compact cover of the IASI measured plume, is as representative as a single injection position (an approach that can be extensively found in literature). Nothing in nature appears in the shape of a box. Therefore, this can only be seen as a **statistical approach**. The trajectories were chosen to show the general transport pathway from the plume region.

-We acknowledge that we do not account for the rising effect of self-radiative heating of the plume. However, taking this effect into account would imply a good characterization of the aerosol optical properties, which is not available from observations. Instead, we launch parcels over a rather thick layer to test the sensitivity of vertical dispersion due to heating. As a matter of fact, once the plume has reached planetary scale dispersion which is attained by the end of August, the subsequent dispersion is mainly barotropic in nature. To support this statement, we added another CLaMS Figure (as seen in Figure 3) at 400K in the supplements (Figure D1).

-The statement 'We know from Khaykin et al. that the initial plume was below 380 K' is wrong, as there are no measurements shown in Khaykin et al. before the 16[th] of August.

-MF argues that it is impossible to have seen increased CO already at the 12[th] of Aug: As implied in the text, we do see a strong local enhancement just above the location of the fire of CO on the 12[th] of August with IASI. This is an undisputable fact (see Figure R1, which we have also added to the new manuscript version). Therefore, we do not see a reason to delay the launch of the simulations beyond the observational evidence of the injection of the plume. Additionally, Peterson et al. (2018) also discuss the first stratospheric appearance of the plume aerosols on the 12[th] of August.

-In Yu et al. it was found that self-radiative heating was necessary for the plume to reach the tropics, but this was obtained in a free running model that was not nudged after the 12[th] of August 2017. In such circumstances, predictability studies tell us that the circulation can diverge completely within ten days with the result that the plume can take a different route than observed. In our simulation the plume was advected using analyzed wind and we claim that it can reach the tropics without the need of additional heating.

Changes to the manuscript:

[revised manuscript text omitted]

**FIVE)**

As Fairlie et al. (2014) give a link between the Nabro volcanic plume and the Asian monsoon circulation, we replace the Bourassa statement.

*'Conditions of enhanced aerosol concentration such as volcanic eruptions can act as tracers of AMA dynamics . For the Nabro volcano eruption, for example, the emitted aerosol and precursors have been partly injected directly into the lower stratosphere (Vernier et al., 2013; Fromm et al., 2013) at altitudes of about 15-18 km (Clarisse et al., 2014; Fromm et al., 2014). It has been shown that the Asian monsoon anticyclone is a dominant feature governing the dispersion of the volcanic plume (Fairlie et al., 2014).'*

**SIX)** The concerns MF expresses here is criticism on the general use of satellite-based aerosol extinction data and the cloud top algorithm for OMPS.

For the plume detection, as stated, we study the plume originating from the Canadian fires already reaching the region within the stratosphere. We do not expect problematic cloud signatures interfering at these levels.

For the signal attributed to the ATAL, although there may be still some debate in the scientific community about the fact that the ATAL detected by the CALIOP space-borne instrument may actually

be a cloud signature, the balloon-borne in situ OPC measurements shown in Vernier et al. (BAMS, 2018) and more recent campaigns in India during the monsoon season (still unpublished results but presented at various international conferences) reveal the presence of particles typically at the altitudes of what is derived from SAGEIII and OMPS. Such particles are by far dominantly small (i.e. smaller than 0.5 μm) and not really what is expected for cirrus clouds. Moreover, it should be noted that these balloon data did not reveal any systematic and ubiquitous presence of high-altitude clouds above India. This should be the case also for the satellite profiles.

We do not agree with MF that the choice of unfiltered cloud signatures in SAGEIII is an oversimplification. Extinction for cirrus clouds can easily be found in the literature and have been shown to reach much stronger values than the ATAL-attributed signals presented in our study, i.e. by a factor of ~10 (see for instance Wang et al., Atmospheric Research, 1994 for SAGEII data similar to SAGEIII in terms of retrieval technique and vertical resolution). None of the SAGEIII profiles within the transport barrier and for the ATAL analysis (Figure 2) show such strong extinction signals at the expected altitude levels for the ATAL. For the current state of knowledge of the microphysics in the monsoon anticyclone, the ATAL aerosols are the best candidates to explain the low extinction signal in SAGEIII observations (below the fire plume layer).

Finally, about MF's objection to the use of the cloud-filtered OMPS dataset we would say that this is the most sophisticated cloud filtering suggested by the OMPS team (by using the cloud-top data), This is the best effort that can be done from a UV-visible spectrometer, by a recognized team. Screening SAGE III data for clouds using two wavelength ratios would be possible. However, whether the cloud screening is perfect or there are still some cloud residuals doesn't change the main argument, that the ATAL signature in the two measurements is real and most of the measurements relevant to this study are in the stratosphere, above the clouds.

As a result, we believe that a signature of an aerosol signal attributed to the ATAL and derived from cloud-unfiltered SAGE III individual profiles and cloud-filtered OMPS data remains highly plausible.

**SEVEN)** We agree with MF that, scientifically the new added lines do not give more scientific content but **have been asked for** as a mean to '**guiding the eye-line'**. In their purpose of guiding the eye and underlining the content of the text, we believe that they fulfill their purpose and make life easier for the reader. We also – of course – strongly agree that one single line cannot represent the complexity of sedimentation etc.. However, we want to emphasize that the respective (see below) sentences in the manuscript do under no circumstances express this. We added more explanation for clarification:
*'Between November 2017 and March 2018, the aerosol signature descends with roughly 0.64 mm in altitude per second based on aerosol extinction values > 0.8 km$^{-1}$ (5 km in three months, October to Janurary). This is in the order of the rate expected for the downwelling of the BDC (see Abalos et al.(2015)). However, other processes may contribute to the visible aerosol extinction decrease in Fig 1B: the effect of sedimentation is expected to play an important role.  The contribution of sedimentation as well as dilution/mixing is not quantified here, microphysical and dynamical sensitivity studies would be necessary. The  lower stratosphere is filled with enhanced aerosols until about mid-April 2018.'* and *'In the tropics, the fire plume signature rises about 0.2-0.3 mm per second (about 5 km from September to April) in the magenta box according to aerosol extinction values of around 0.6 km$^{-1}$. This tropical upwelling velocity estimate is in good agreement with the tropical upwelling velocity in current reanalyses (e.g. Abalos et al. (2015), Fig. 6). Similar ascending features are visible around the globe 0-25°N. The reversed vertical transport of the aerosol particles in Fig. 1B*

*compared to 1D (i.e. the observed descent in the northern latitudes and ascent in the tropics) reflects the contribution of the ascending and descending branch of the BDC. The average signal for the magenta box remains also until April 2018 at ~19 km altitude. Based on the good agreement of the slope in the aerosol signal with downwelling and upwelling velocities by the BDC we hypothesize that the BDC played a role in both extratropical downward and tropical upward transport of the aerosol.*'

**SEVEN)** Other than suggested by MF there is no discrepancy between the results in Khaykin et al. 2018 and this study.: The self- rising features Khaykin et al. (and Yu et al., 2019) are discussing, happen on a much shorter time scale (less than 2 months) and can therefore not be implemented for any longer-term analysis as done in this study.
We already mentioned that the motion to the tropics does not necessarily need the self-rising process. Please also refer to the reply to 'Similar ascending features are visible around the globe 0-25°N-issue' (below) and the given Figure within.

**EIGHT)** The respective paragraph in the previous version of the manuscript explains this: paragraph starting at the end of page 6 (now end of page 7). Tropospheric cloud signatures amongst mid-latitude fire-aerosol signatures are extremely dominant and saturate the scale (to dark red), also when averaged with other profiles. Therefore, we have focused on a specific aerosol extinction range **(see also SIX)**.
For the respective sentence we removed 'troposphere' as we focus on the mid-latitude stratosphere.
'*The  lower stratosphere is filled with enhanced aerosols until about mid-April 2018.*'

**NINE)** As MF has noticed later in his comments (**TWELVE**) the Figure is found in the supporting material (Fig B1).
Previously MF has argued that the reason for adding this plot to the manuscript has not sufficiently been explained, which is why we have added more explanations (in the last revised manuscript version) in the text to why this plot is of big asset to this study.
For clarification of the current misunderstanding we have added some cross referencing: '*The unfiltered cloud structures in the SAGE III data set masks the first appearance of the plume in the black box in Fig. 1C. However, the first SAGE III profile in the Asian monsoon region, that we can trace back to the fire plume signature  and that has previously been transported within the circulation of the AMA appears on August 30th 2017 at 17 km altitude (Fig. B1, within the green box). As a result, there is no indication of a fire plume signature within the AMA core (i.e. passing the AMA transport barrier). The relatively high altitudes of this signature (17-20 km) in Figure 1C indicate that the fire plume arrived in the TTL region in the Asian monsoon area, where the transport barrier  might have forced the fire plume to rise by bypassing the AMA on its upper part, as it was the case for the Sarychev aerosol plume in 2009 (Vernier et al., 2011a).*'

**TEN)** 1st As before we agree that the line does not represent any new scientific content (see also **SEVEN**).
2nd The 'image' of an ascent discussed in Khaykin et al. happens in the order of days (they look at ~2 weeks), not in the order of months. The self-rising feature that Yu et al. 2019 discuss, happens in a time frame of 2 months (here we look at 7 months). The role of the self-rising effect has already been discussed in **FOUR**.

**ELEVEN)**

The AMA is the main driver of extra-tropics- tropics exchange in the summer in the UTLS. This is the topic of our work. We do recognize that there are other mechanisms possible, therefore we have added two new paragraphs (see below):

We have added a small paragraph in the introduction to introduce the literature: *'A few rare extreme-fire events prior to this case have been investigated in terms of global distribution, showing enhanced trace gas and aerosol signatures in tropical latitudes (e.g. Siddaway&Petelina 2011, Pumphrey et al. 2011, Jost et al. 2004 and Fromm et al. 2008). Those studies primarily focus on the evolution and distribution of the respective fire plumes rather than the underlying transport processes.'*

Additionally, to provide a global view of the plume transport characteristics, we have added a whole new paragraph in the conclusions, discussing/ mentioning other processes that can potentially impact the transport of the plume in the northern hemisphere and are not considered in this study:
*'Most of the Canadian fire aerosols injected into the northern hemisphere stratosphere have been transported to the north, descending back to the troposphere via the lower branch of the BDC (Yu et al.,2019). In this study, we analyze one specific southward transport path way: the transport in the lower stratosphere within the easterly jet together with the north-south anticyclonic flow at the eastern flank of the AMA. We show that this is an efficient transport pathway from northern latitudes to the tropics. However, other dynamical processes bringing air masses from the mid-latitudes to the tropics are possible and not investigated in this study: for instance, the easterly jet in the lower stratosphere does not appear only at one specific latitude, but can also allow for a southward (also northward) propagation of air masses. Furthermore, fire plume air masses injected in the mid-latitude upper troposphere to the tropics with subsequent uplift to the stratosphere could be considered. Beside the AMA, which is the by far largest periodically reoccurring anticyclonic flow system in the lower stratosphere on Earth, dynamical effects of other monsoon systems have the potential of transporting air masses southwards (e.g. the North-American monsoon and the West-African monsoon) in the lower stratosphere.'*

**TWELVE)**
Please see similarity to point **NINE**. This comment has already been answered.

**Other 'major' comments**

**"Whole >40°N" -issue:**
As we are showing a time frame of more than one year and we use all measurements north of 40°N (including measurements up to 70°N), calling it >40°N is not wrong. However, we agree that the wording of 'whole >40°N' can be improved. We have made changes to the text: *'A strongly enhanced aerosol extinction signature appears in the SAGE III data set,  >40°N mid to end of August (Fig. 1B), after the beginning of the major fire event in Canada, confirming the results of multiple previous studies (Khaykin et al., 2018; Ansmann et al., 2018; Haarig et al., 2018). It has to be noted that the measurement point furthest north was taken below 70°N, however, especially in the winter most observations are limited to below 50°N.'*
and the Figure caption: *'(B) all longitudes and latitudes* available *above 40°N with an average of 88 profiles per bin (however, never higher than 70°N),'*

**"Explanation of the aerosol extinction decrease in Fig 1B"- issue:**

Nowhere in the text do we state or imply that the BDC is the only driver of the visible decrease. However, the numbers match extremely well and therefore we still think that the role of the BDC is crucial even in the observed narrow 40-50°N belt during winter months. As also seen in the CLaMS simulation, already in September the fire tracer- air masses are well distributed throughout the northern hemisphere. The mentioning of dilution and mixing is therefore more appropriate than a simple transport of plume-air masses in and out of the '10-degree belt'. Therefore, it is justified to use an 'only' 10-degree wide band to infer the trend (one must also note that the decadal aerosol trends have been reported from one single site, i.e. Laramie; the same for water vapor in Boulder). Interestingly, we note that Figure 1A from Yu et al. (2019) also shows some decay of the plume altitude at mid-latitude (15-60°N in their case) throughout time with a timing very similar to what is shown on our Fig 1B. The respective paragraph has been changed **(see SEVEN).**

**"Fig 1D upwelling in small box"- issue:**
See also **SEVEN**.

1) The sentence in the manuscript *'Similar ascending features are visible around the globe 0-25°N'* should account for this issue (including both MF simulations). For the issue with this sentence itself, see the next point.

2) Other studies also focus on analysis of stratospheric dynamical features based on fixed latitudes and specific longitudes (see for instance Demirhan Bari et al., The effect of zonal asymmetries in the Brewer-Dobson circulation on ozone and water vapor distributions in the northern middle atmosphere, JGR, 2013). Also, as pointed out in our reply to the previous comment, dynamical variability in the stratosphere (aerosols and water vapor) have been commonly derived from restricted areas. So, our analysis cannot be considered as uncommon.

This said, the interest of showing the fire plume aerosol variability in the magenta box which is an extension of the black box to tropical latitudes is twofold: 1) to point out the differences in terms of aerosol loading with the more isolated core of the AMA at the end of the monsoon season (Figure 2 mostly), and 2) to extend our analysis after the monsoon season for the same regions (defined by the black and magenta boxes) with a box which is largely influenced by mid-latitude dynamics (the black one) and another box which encompasses the dynamical effects of the tropics (the magenta one). In other words, after the break-up of the AMA the 2 boxes are maintained for continuity on the Figures 1C-D but each box is still representative of the 2 different stratospheric compartments. Then, on Figure 1C-D we do see different trends with some increase in the magenta box as influenced by tropical dynamics and the BDC ascending branch (as we mention in the text, this influence was pointed out by Vernier et al. (2011b) for volcanic eruptions). As stated in the text, the conclusions are the same if we take a 0-25°N zonal mean (see reply to next comment).

**"Similar ascending features are visible around the globe 0-25°N"-issue:**
We would like to divide this comment by MF in two parts:
- **1) MF: Doubt about whether this statement is true and if it is, it should appear as a supplement:**
  This is the mentioned plot. The ascent is clearly visible. Originally this is a plot we made for ourselves to even be able to write a sentence like this (we find it highly unusual to be doubted for trustworthiness for a statement like this), but we added this plot as supporting material.

[Figure]

*Figure R3*

*Figure caption C1: Same as for Fig 1D, but considering all measurements from 0-25°N.*
*Supporting material for Figure 1D: As the upwelling of the BDC in the tropics is a feature usually displayed on a bigger scale, a larger area covering all longitudes in the northern hemispheric tropics (0-25°N) was chosen (Fig. C1). The ascending feature of enhanced aerosol extinction values (as seen in Fig. 1D) are clearly visible in Fig. C1 and can therefore be attributed to the rising branch in the tropics of the BDC.*

- **2) Even if true, other mechanisms like the self-rising are important and should be mentioned: As stated before** the self-rising feature (diabatic lofting of the plume), which is a very important process for this plume, has no significant impact on long term trends as presented in this work, but rather on shorter scales (weeks-2 months) (Please also see answer to **SEVEN**). Other significant, ascending processes that are only visible in the tropics and are in the order of the upwelling of the BDC are unknown to us.

**"Why is the ATAL mentioned at all (discussion to Fig 1E)"-issue :**
The explicit description and motivation of how the fire plume and the ATAL are analyzed together in one paper **is given in the introduction** (please refer to the introduction for the motivations for this study). One point of the paper is to quantitatively compare some effects of the ATAL and the fire plume (please also see Figure 2).
The suggestion by MF to remove the respective paragraph would remove all of the description for Fig. 1E, which we find confusing. MF claims that the connection between the ATAL and the fire signal is not clearly explained: In our opinion it is. The paragraph clearly states that we do **not** see a clear connection between both events (both driven by AMA related dynamics). That is an important outcome.

**"Fire plume vs. confinement"- issue:**
We are not completely sure about the reason of confusion brought forward by MF. We assume that he has not completely understood that the level of confinement depends on the altitude, therefore we have added more explanation in the text:
*'…The fire plume appears at altitudes between 16 and 21 km at the beginning of September 2017 and is not visible anymore at the end of March 2018. Both signals appear as two distinguishable events on different altitude ranges (fire plume: 16-22 km, ATAL: <18 km). In Fig. 1E, the ATAL is mostly apparent in the troposphere (at altitude levels exhibiting a generally higher AMA confinement but as expected also with some distinguishable signal in the lowermost stratosphere), while the fire plume appears clearly in*

*the stratosphere (at altitude levels with generally weaker confinement), which indicates a clear separation between both aerosol layers....'*

For individual profile(s) (later in the manuscript, discussion to Fig 5), the strength of the transport barrier at different potential temperature levels is further explained.

**"Appendix B"- issues:**

Rather than P6L08-L31, we believe MF is referring to L27-31 as he is exclusively talking about Figure B2 (B1 in the current version):

**As already stated in the manuscript** (see the respective paragraph below), the difference between the end points of the blue and red trajectories is exactly the argument we make. We do not argue that the trajectories (initialized from different altitude levels) have taken completely different paths, but emphasize that the ones that we attribute to the fire signature get much closer to areas where fire signature has already been observed (and published in Khaykin et al.,2018, please refer to Figure 3). To clarify we have made some changed to the respective paragraph:

*'While the 9 day back trajectories from the peak at 17 km altitude (red trajectories, reach out to western* Asia, *the blue trajectories, corresponding to the lower peak at 15 km do not reach as far to the west. At this time the* very *first fire plume signal is observed in western Asia, then transported within the next week over central and east Asia (Khaykin et al., 2018 in Fig. 3). Consequently, the higher peak at 17 km altitude (with air mass influence from far western Asia on the 21st of August) may be interpreted as a mixture of air masses partly originating from the Canadian fire plume. The lower peak at 15 km altitude (with 9- day backtrajectories reaching back to regions where the fire plume has not been transported to yet and staying closer to the center of the AMA) can be associated with the ATAL signal (Fig. B1).'*

**"AppendixB-false referencing"-issue:**

The sentence has been removed:

**'Minor' Comments in the manuscript PDF:**

Smaller corrections (which are not further discussed in the following) regarding spelling/missing symbols are directly added to the text.

**MF: This reader is unaware of any paper(s) that support this broad assertion. Please provide applicable citations** = citations for the impact of the Asian monsoon on the TTL

We added various citations supporting this statement. However, many more are possible to cite. The citation already given in the respective sentence fits well.

*'The Asian summer monsoon influences the composition of the Upper-Troposphere–Lower-Stratosphere (UTLS) (Garny and Randel (2016), Pan et al. (2016) and Ploeger et al. (2017)), especially in the Tropical Tropopause Layer (TTL: as defined in Fueglistaler et al. (2009), Fig. 5a).'*

**MF: How does the extinction range exclude clouds? Can't clouds have such extinctions? (e.g. SVC)**

The presence of sub-visible clouds cannot be excluded but is expected to have a minor impact on the extinction.

(See also **SIX** and **EIGHT**)

**MF: How does this "confirm" the results of these other papers?**

We have added more explanation accordingly:

*'A strongly enhanced aerosol extinction signature appears in the SAGE III data set, in the whole NH >40°N mid to end of August (Fig. 1B), after the beginning of the major fire event in Canada. This confirms the results of multiple previous studies, which have also seen highly increased aerosol signatures due to the fire mid to end of August at higher latitudes in the NH (Khaykin et al., 2018; Ansmann et al., 2018; Haarig et al., 2018).'*

**MF: The two sentences juxtaposed here make no sense to me. The first ("The average signal...") refers to the winter half of the year, the second ("The AMA generates...") refers to the summer season. Please clarify the argumentation.**
We have placed the second sentence further to the front:
*'To see whether the fire plume has  been transported to the tropics (as it has been shown for the Sarychev eruption by Wu et al. (2017)), another box south of the core Asian monsoon box has been chosen (Fig. 1A, magenta box). We attribute the ascending signal starting at around 16 km in mid-September and reaching altitudes of around 21 km about 6 months later to the Canadian wildfire, as its origin coincides in time and altitude with the fire signal in the AMA region (black box, Fig. 1C). Hence, the AMA generates a strong connection between the mid-latitudes and the tropics during the summer season.'*

**MF: It's not evident what this sentence has to do with the prior ones in this paragraph. The reader needs to know how this statement supports the text leading into it.**
*'To further verify the link between the enhanced aerosol extinction values in the Asian monsoon region (especially south of the center, as seen in Fig. 1) and the Canadian wild fire plume, individual SAGE III observations are further analyzed in Fig. 5 and B1 to B3.'*

**MF: K19 have stipulated that they are only following stratospheric aerosol. What is the connection to the AT(ropopause)AL?**
There are both aerosol enhancement phenomena in the UTLS. Please refer back to the introduction.

**MF: By definition the ATAL is tropospheric. Why does this need to be stated?**
Answer: This is a description of Figure 1E. For further explanation, see text.
Furthermore, the ATAL is not tropospheric by definition (tropical tropopause is a vertically-extended transitional zone between the tropical troposphere and stratosphere, see e.g. Randel and Jensen, Nature Geoscience, 6, pages 169–176 (2013)).

**MF: What is meant by "zonal"? This doesn't look like a mapping of zonal wind.**
'zonal' has been removed:

[revised manuscript text omitted]